# Transcriptional networks specifying homeostatic and inflammatory programs of gene expression in human aortic endothelial cells

Nicholas T Hogan[1], Michael B Whalen[2], Lindsey K Stolze[2], Nizar K Hadeli[2], Michael T Lam[1], James R Springstead[3], Christopher K Glass[1], Casey E Romanoski[2]*

[1]Department of Cellular and Molecular Medicine, University of California, San Diego, San Diego, United States; [2]Department of Cellular and Molecular Medicine, University of Arizona, Tucson, United States; [3]Department of Chemical and Paper Engineering, University of Western Michigan, Kalamazoo, United States

*For correspondence:
cromanoski@email.arizona.edu

**Abstract** Endothelial cells (ECs) are critical determinants of vascular homeostasis and inflammation, but transcriptional mechanisms specifying their identities and functional states remain poorly understood. Here, we report a genome-wide assessment of regulatory landscapes of primary human aortic endothelial cells (HAECs) under basal and activated conditions, enabling inference of transcription factor networks that direct homeostatic and pro-inflammatory programs. We demonstrate that 43% of detected enhancers are EC-specific and contain SNPs associated to cardiovascular disease and hypertension. We provide evidence that AP1, ETS, and GATA transcription factors play key roles in HAEC transcription by co-binding enhancers associated with EC-specific genes. We further demonstrate that exposure of HAECs to oxidized phospholipids or pro-inflammatory cytokines results in signal-specific alterations in enhancer landscapes and associate with coordinated binding of CEBPD, IRF1, and NFκB. Collectively, these findings identify cis-regulatory elements and corresponding trans-acting factors that contribute to EC identity and their specific responses to pro-inflammatory stimuli.

## Introduction

Atherosclerosis is an inflammatory disease of large arteries mediated by the accumulation of plaque within the vessel wall. Through sequelae such as heart attack, stroke, and peripheral vascular disease, it is responsible for an immense burden of morbidity and mortality. The pathogenesis of atherosclerosis involves several cell types and environmental risk factors (*Lusis, 2000*; *Glass and Witztum, 2001*). One of the critical cell types is the arterial endothelial cell (EC). The onset of atherosclerosis involves the activation of ECs by pro-inflammatory micro-environmental exposures including hemodynamic turbulence, oxidized-specific epitopes, and inflammatory cytokines (*Tabas et al., 2015*). These inflammatory stimuli result in the expression of adhesion molecules on the luminal EC surface and rolling, attachment, and migration of leukocytes into the vessel wall. Sustained recruitment and accumulation of immune cells in the vessel wall leads to extracellular matrix remodeling, smooth muscle cell migration, and the development of necrotic debris. Acute plaque rupture can result in sudden vascular occlusion, leading to heart attack or stroke.

Genome-wide association studies have identified more than 50 loci that predispose humans to cardiovascular disease (CVD) (*Nikpay et al., 2015*), of which the major cause is atherosclerosis. The

majority of CVD loci reside outside protein-coding regions of the genome, suggesting that the risk variants alter gene regulatory function (*Hindorff et al., 2009*; *Manolio, 2010*). Still, the target genes, pathways, and cell types of action are largely unknown due to challenges in linking regulatory variants to function. A major challenge is that mammalian genomes contain upwards of a million potential regulatory elements called *enhancers*, yet a given cell type only utilizes on the order of tens of thousands of active enhancers (*ENCODE Project Consortium, 2012*; *Andersson et al., 2014*). This makes it difficult to accurately predict the functional cell systems and units of regulation from sequence alone (*Shlyueva et al., 2014*).

An important insight into enhancer biology is the observation that unique combinations of a few transcription factors (TFs) together activate cell-type-specific enhancers. Enhancer priming by TFs is both collaborative, (such that one TF will not bind its DNA motif if the motif for a collaborating TF is mutated [*Heinz et al., 2013*]), and hierarchical (the majority of sites bound by newly abundant TFs occur at enhancers pre-bound by collaborating TFs [*Heinz et al., 2015*; *Romanoski et al., 2015*; *Kaikkonen et al., 2013*]). This model is perhaps best characterized in the hematopoietic system and with toll-like receptor 4 signaling (*Heinz et al., 2013*; *Kaikkonen et al., 2013*; *Heinz et al., 2010*). For example, myeloid-specific enhancer activation and cell differentiation requires the TF PU.1 in combination with C/EBPb, whereas B cells require PU.1 in combination with EBF and E2A (*Heinz et al., 2010*).

In the current study, we take a genome-wide approach using DNA variation, epigenetic, and transcriptomic data to identify the major TF families that coordinate human aortic endothelial cell (HAEC) gene expression in homeostasis and upon exposure to prototypic inflammatory stimuli characteristic of atherosclerosis. Using a combination of experimental and computational approaches, we find that members of the ETS and AP1 TF families bind EC enhancers and that removing ETS member ERG elicits an inflammatory profile. We demonstrate that many enhancers identified in ECs are cell type-specific and several enhancers overlap with SNPs that have been associated to coronary artery disease (CAD) and hypertension. In addition, we demonstrate that TFs NRF2, NFκB, CEBD, and IRF1 are signal-dependent TFs that mediate the EC response to inflammatory stimuli.

## Results

### Transcription factors in the AP1 and ETS families dominate the enhancer landscape in HAECs

A total of 16,929 high-confidence enhancer-like elements were mapped in HAECs (*Figure 1a*) using chromatin immunoprecipitation followed by high-throughput sequencing (ChIP-seq) to identify promoter-distal elements marked by significant levels of histone H3 di-methylation of lysine 4 (H3K4me2) and acetylation of lysine 27 (H3K27ac) that together mark active enhancers (*Heintzman et al., 2007*; *Creyghton et al., 2010*; *Rada-Iglesias et al., 2011*). Chromatin accessibility, measured by Assay for Transposase Accessible Chromatin with high-throughput sequencing (ATAC-seq [*Buenrostro et al., 2013*]), was used to center the enhancer-like regions. The position within the element with the maximum signal that reflects greatest accessibility was used for centering. Using public global nuclear run-on sequencing (GRO-seq) data in HAECs (*Kaikkonen et al., 2014*), we observed that our set of enhancer-like loci produced bi-directional nascent RNA transcripts, or enhancer RNAs (eRNAs), as evidenced by the red and blue strand-specific RNA signals in *Figure 1a*. The potential function of eRNAs is not understood; however, eRNA output is robustly correlated with enhancer activity (*Kaikkonen et al., 2014*; *Lam et al., 2013*; *De Santa et al., 2010*; *Kim et al., 2010*), further supporting our enhancer set as active EC enhancers (*Figure 1a*).

We hypothesized that the major TFs that select and maintain enhancers in HAECs would be evident via enrichment of binding motifs in enhancer DNA sequences. Thus, we performed de novo motif enrichment analysis and discovered that AP1, ETS, SOX and GATA motifs were significantly enriched (-logPvalues > 7.1e2) in HAEC enhancers compared to random GC-matched genomic background sequence (*Figure 1b*, comprehensive list in *Figure 1—figure supplement 1*). Based on previous evidence (*Heinz et al., 2013*), we expected for functional motifs to be enriched near the maximum signal for chromatin accessibility. Indeed, AP1 and ETS were most frequently observed near the signal maximum, whereas the relationships for SOX and GATA motifs were less pronounced

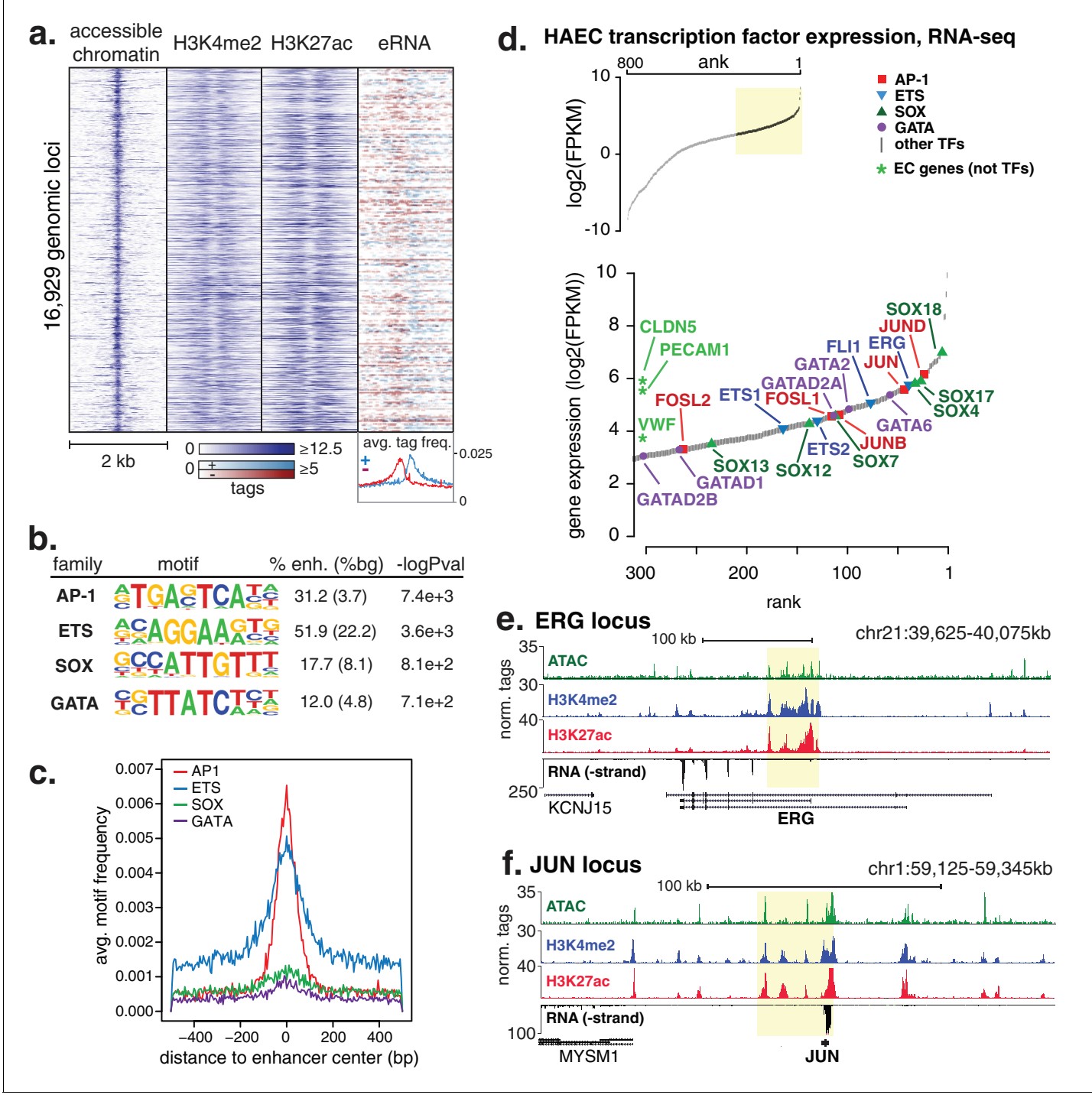

**Figure 1.** HAECs display a distinct repertoire of enhancers that nominate combinations of the AP1, ETS, SOX and GATA TF families as major orchestrators of HAEC gene expression. (a) A heatmap of 16,929 enhancer-like regions were selected by: accessible chromatin (ATAC-seq), coincident with H3K4me2 and H3K27ac deposition (ChIP-seq) in gene-distal positions (≥3 kb from promoters). Rows are enhancer loci, repeated for each data type in columns. Bidirectional transcription of enhancer RNAs (eRNAs) is also evident (GRO-seq). (b) The top four enriched motifs that occur in enhancers from a are shown. The transcription factor (TF) family, de novo motif matrix, percentage motif occurrence at enhancer loci versus random loci, and enrichment -logPvalues are indicated. Enrichment was calculated from 200 bp sequence, centered on chromatin accessibility. (c) The positional enrichment of the enriched motifs are shown relative to the center of the enhancer-like elements from a, where 0 bp is the center of the accessibility signal. (d) Gene expression measured by RNA-seq and limited to TFs is ranked by FPKM values to nominate the most highly expressed TFs in the AP1, ETS, SOX, and GATA families. (e, f) RNA expression, histone modifications, and super enhancer (SE) definitions are shown for: (e) ERG and (f) JUN loci. SE regions are highlighted in yellow and were defined using H3K27ac. More related data in *Figure 1—figure supplements 1–3*.
*Figure 1 continued on next page*

*Figure 1 continued*

The following figure supplements are available for figure 1:

**Figure supplement 1.** Analysis of motifs and expression of associated transcription factors.

**Figure supplement 2.** Genetic loci for JUN, JUNB, and JUND.

**Figure supplement 3.** Hierarchical clustering of enhancers.

(*Figure 1c*). These data nominate roles for AP1 and ETS TF members as important mediators of the HAEC enhancer landscape.

Tens of genes encode proteins of the AP1, ETS, SOX and GATA TF families. Within each family different members share nearly identical DNA-binding domains and thus bind the same motif. In addition, AP1 protein members bind AP1 motifs as homo- and hetero-dimers. These sources of redundancy make it challenging to identify the functional family member(s) without additional information. To narrow the search in HAECs, we hypothesized that the operational TFs would be highly expressed and that their genetic loci would contain super-enhancers (SEs), or unusually dense clusters of highly decorated enhancers (*Hnisz et al., 2013*). Characterization of enhancer marks across cell-types has found SEs to be frequently located at loci encoding lineage-defining TFs (*Whyte et al., 2013*; *Dowen et al., 2014*). We queried rank-ordered expression data for TF family members and found that multiple members of each group were highly expressed (*Figure 1d*). For example, SOX members SOX18, SOX4, and SOX17 were among the top 4% most expressed TFs in HAECs. AP1 family members JUND, JUN, and JUNB were also in the top 4%. RNA-seq from other HAEC donors and replicate samples confirmed these findings (*Figure 1—figure supplement 1*). Next, we defined SEs using H3K27ac ChIP-seq data and found that, among others, the genetic loci for ERG (an ETS member) as well as AP1 members JUN, JUND, and JUNB harbored SEs (*Figure 1e, f*, *Figure 1—figure supplement 2*). Taken together, these data suggest that while multiple TFs from each family probably bind HAEC enhancers, that JUN, JUNB, JUND, and ERG likely serve prominent roles.

## Roughly half of HAEC enhancers are endothelial-specific

To investigate which enhancer-like elements discovered in HAECs were specific to ECs, we analyzed public H3K27ac ChIP-seq datasets from ENCODE (*ENCODE Project Consortium, 2012*) and Roadmap Epigenomics consortia (*Kundaje et al., 2015*). Considering ECs are present in nearly all tissues, we focused on data collected in single cell types with the exception of 'aorta', 'right ventricle', 'left ventricle', and 'right atrium' that were included to observe their relationship to aortic endothelium. A total of 61 datasets were analyzed (*Supplementary file 1*). Human umbilical vein ECs, or HUVECs, were the only other EC type in the analysis. H3K27ac ChIP-seq tags were counted and normalized in each experiment at the 16,929 HAEC-defined enhancer loci. Hierarchical clustering resolved three distinct clusters of enhancers: an endothelial-specific set (n = 7405), a set common across cell types (1575), and a mixed set where only some cell types exhibited H3K27ac modification (7949) (*Figure 1—figure supplement 3*). Motif analysis of these three sets revealed differential frequencies of AP1, ETS, SOX, and GATA motifs (*Figure 1—figure supplement 3*). AP1 and ETS motifs were least frequently observed in the common enhancer set, while the ETS, GATA, and SOX motifs were most frequently observed in the endothelial-specific enhancer set. These data are consistent with the model that different combinations of transcription factors maintain cell-specific gene expression programs.

## Aortic endothelial enhancers overlap genome-wide association SNPs for CAD and hypertension

To investigate whether EC enhancers have utility to prioritize non-coding functional variants for the cardiovascular diseases CAD and hypertension, we overlapped physical coordinates of the 16,929 enhancers from *Figure 1a* with GWAS associated variants. SNPs meeting genome-wide significance

for CAD or hypertension, which is a major risk factor for atherosclerosis and CAD, were downloaded from the NHGRI-EBI GWAS Catalog (*Welter et al., 2014*). To account for linkage disequilibrium (LD, the correlation of alleles) between closely spaced SNPs on the same chromosome, we used 1000 Genomes data (*Auton et al., 2015*) to retrieve SNPs in LD with the reported GWAS SNPs when r2 was greater than 0.8 based on European haplotype structure. We identified 16 SNPs that were within HAEC enhancers (*Table 1*) and represent 22 lead SNPs from GWAS studies. Fifty percent of overlapping SNPs were within EC-specific enhancers (as opposed to those common or mixed across cell types), whereas only 43% of enhancers in HAECs are EC-specific (*Figure 1—figure supplement 3*). These data provide a focused list of potential functional non-coding variants that affect predisposition to CAD and hypertension through EC gene regulation. Further studies will be required to establish the regulatory consequence and predisposing mechanisms of these variants. Nonetheless, our evidence that perturbed endothelial expression contributes to vascular disease underscores the importance of elucidating endothelial gene regulatory programs in homeostasis and inflammatory environments.

## TF expression dynamics across 97 HAEC donors nominates three major modules of TFs as coordinating gene expression

We next questioned how AP1, ETS, SOX, and GATA TFs were expressed in artery ECs across the human population. We postulated that the most prominent actors would be highly expressed with modest variation between people. By leveraging global transcript levels collected across 97 genetically distinct HAECs from healthy human donors (*Romanoski et al., 2010*), we found that JUN and JUND (AP1) and ERG (ETS) exhibited the greatest median expression values with relatively little variability across the EC donor population (*Figure 2a*).

To gain insight into the behavior of the TF members with respect to each other, we measured co-variation in TF gene expression profiles across the human population. Co-variation, or co-expression, of TFs could result from one TF (in)directly regulating another, both (in)directly regulating each other, or from each being regulated by a common third mechanism. By clustering pair-wide correlation coefficients across all TFs of interest, we identified three main groups with similar co-expressed profiles: group 1 (in orange) with members FOSL1 and ETS1; group 2 (in green) with GATA2 and GATA6; and group 3 (in yellow) with the remaining factors (*Figure 2b*, detailed examples in *Figure 2c-f*). The degree of correlation between TFs is indicated by red intensity (anti-correlation with blue; no correlation with white). Notably, TF expression of groups 1 and 2 were mostly anti-correlated with group three members. A very similar grouping of these TFs was observed when their relationship to all expressed genes was used as the clustering parameter (*Figure 2—figure supplement 1*). The result that FOSL1 and ETS1 are anti-correlated in expression with the remaining family members, and to HAEC transcripts overall, suggests that they promote opposing gene expression profiles in HAECs.

## Nominated factors, including ERG and JUN, bind HAEC enhancers at closely spaced motifs

To test whether the nominated TFs indeed bound HAEC enhancers, we performed the first chromatin immunoprecipitation sequencing (ChIP-seq) experiments for JUNB, JUN, and ERG in HAECs and analyzed GATA2(*ENCODE Project Consortium, 2012*) and ETS1 (*Zhang et al., 2013*) binding data from human umbilical vein endothelial cells (HUVECs). The JUND cistrome would also be informative in these studies; however, we proceeded with JUN and JUNB because the heterodimeric binding of AP1 factors makes it likely that JUN and JUNB profiles encompass a major portion of the overall AP1 landscape. JUN, JUNB, ETS1, and GATA2 were all confirmed to bind active HAEC enhancers in the open chromatin region (*Figure 2—figure supplement 2*). As determined by clustering of binding profiles, JUNB and JUN were similar and ERG and ETS1 were similar, supporting the role of canonical DNA motifs on factor recruitment. Next, we asked if there was enrichment of other motifs proximal to the bound motifs as was observed previously for TF pairs known to collaboratively activate cell-specific enhancers (*Heinz et al., 2010*; *Gosselin et al., 2014*). For this analysis, loci for the bound factor (e.g. ERG) were centered on the respective motif (e.g. ETS motif) and sites lacking the motif were omitted. Then, frequencies of other motifs were calculated as a function of distance to the reference motif. We found that AP1 motifs were most frequently oriented within 50 base pairs

**Table 1.** Overlap of HAEC enhancers with GWAS loci reported for coronary artery disease (CAD) or hypertension (HT). Associated SNPs were downloaded from the NHGRI-EBI Catalog of published genome-wide association studies. SNPs in linkage disequilibrium (LD) to GWAS association traits were calculated when r2 >0.8 according to the European reference population of the 1000 Genomes Project. HAEC enhancers defined in **Figure 1a** were overlapped by physical position (hg19 genome build). The GWAS SNP, p-value, GWAS trait, gene reported, PMID, overlapping HAEC enhancer coordinates and enhancer type are shown.

| GWAS SNP | | | | | | HAEC enhancer | | |
|---|---|---|---|---|---|---|---|---|
| SNP in enhancer | LD to lead SNP from study | p-Value of lead | Trait | Reported gene of lead | PubMed ID | Position (chr, start bp, end bp) | Nearest gene | Type |
| rs12091564 | Lead | 2.0E-07 | CAD | HFE2 | 21626137 | 1, 145395579, 145395699 | LOC101928979 | common |
| rs72701850 | LD, rs12091564, r2 = 0.95346 | 2.0E-07 | CAD | HFE2 | 21626137 | 1, 145396840, 145397006 | LOC101928979 | common |
| rs72701850 | LD, rs10218795, r2 = 0.95346 | 2.0E-07 | CAD | HFE2 | 21626137 | 1, 145396840, 145397006 | LOC101928979 | common |
| rs56348932 | LD, rs17114036, r2 = 0.916823 | 4.0E-19 | CAD | PLPP3 | 21378990, 24262325 | 1, 56988477, 56988661 | PLPP3 | EC-specific |
| rs56348932 | LD, rs9970807, r2 = 0.942868 | 2.0E-09 | CAD | PLPP3 | 26343387 | 1, 56988477, 56988661 | PLPP3 | EC-specific |
| rs56348932 | LD, rs17114046, r2 = 0.942868 | 3.0E-07 | CAD | PLPP3 | 21846871, 21378988 | 1, 56988477, 56988661 | PLPP3 | EC-specific |
| rs10047079 | LD, rs2229238, r2 = 0.866848 | 7.0E-07 | CAD | ILR6 | 22319020 | 1, 154468114, 154468189 | SHE | EC-specific |
| rs55916033 | LD, rs10496288, r2 = 1 | 2.0E-09 | HT | intergenic | 21626137 | 2, 83278987, 83279062 | LOC1720 | EC-specific |
| rs55916033 | LD, rs10496289, r2 = 1 | 2.0E-09 | HT | intergenic | 21626137 | 2, 83278987, 83279062 | LOC1720 | EC-specific |
| rs72836880 | LD, rs10496288, r2 = 1 | 2.0E-09 | HT | intergenic | 21626137 | 2, 83308909, 83309314 | LOC1720 | EC-specific |
| rs72836880 | LD, rs10496289, r2 = 1 | 2.0E-09 | HT | intergenic | 21626137 | 2, 83308909, 83309314 | LOC1720 | EC-specific |
| rs112798061 | LD, rs10496289, r2 = 1 | 2.0E-09 | HT | intergenic | 21626137 | 2, 83308909, 83309314 | LOC1720 | EC-specific |
| rs3748861 | LD, rs13420028, r2 = 0.916266 | 1.0E-10 | HT | GPR39 | 21626137 | 2, 133196310, 133196505 | GPR39 | mix |
| rs3748861 | LD, rs10188442, r2 = 0.916266 | 1.0E-10 | HT | GPR39 | 21626137 | 2, 133196310, 133196505 | GPR39 | mix |
| rs144505847 | LD, rs6725887, r2 = 1 | 1.0E-09 | CAD | WDR12 | 21378990, 24262325 | 2, 203672243, 203672412 | ICA1L | EC-specific |
| rs144505847 | LD, rs7582720, r2 = 1 | 3.0E-08 | CAD | WDR12 | 24262325 | 2, 203672243, 203672412 | ICA1L | EC-specific |
| rs56155140 | LD, rs17087335, r2 = 0.979112 | 5.0E-08 | CAD | NOA1, REST | 26343387 | 4, 57824385, 57824541 | NOA1 | mix |
| rs5869162 | LD, rs6452524, r2 = 0.924698 | 2.0E-07 | HT | XRCC4 | 21626137 | 5, 82393827, 82393921 | XRCC4 | EC-specific |
| rs5869162 | LD, rs6887846, r2 = 0.924698 | 2.0E-07 | HT | XRCC4 | 21626137 | 5, 82393827, 82393921 | XRCC4 | EC-specific |
| rs6475604 | LD, rs7865618, r2 = 0.940597 | 2.0E-27 | CAD | MTAP | 21606135 | 9, 22052677, 22052823 | CDKN2B | EC-specific |
| rs17293632 | LD, rs72743461, r2 = 1 | 1.0E-07 | CAD | SMAD3 | 26343387 | 15, 67442510, 67442670 | SMAD3 | common |
| rs17293632 | LD, rs56062135, r2 = 0.988489 | 5.0E-09 | CAD | SMAD3 | 26343387 | 15, 67442510, 67442670 | SMAD3 | common |
| rs17227883 | LD, rs17228212, r2 = 0.981438 | 2.0E-07 | CAD | SMAD3 | 17634449 | 15, 67442769, 67443128 | SMAD3 | common |

*Table 1 continued on next page*

*Table 1 continued*

| GWAS SNP | | | | | | HAEC enhancer | | |
|---|---|---|---|---|---|---|---|---|
| SNP in enhancer | LD to lead SNP from study | p-Value of lead | Trait | Reported gene of lead | PubMed ID | Position (chr, start bp, end bp) | Nearest gene | Type |
| rs1563966 | LD, rs1231206, r2 = 0.844151 | 9.0E-10 | CAD | intergenic | 21378990 | 17, 2095878, 2096222 | LOC101927839 | mix |
| rs1563966 | LD, rs216172, r2 = 0.909315 | 1.0E-09 | CAD | SMG6, SRR | 21378990, 26343387 | 17, 2095878, 2096222 | LOC101927839 | mix |
| rs7408563 | LD, rs7246657, r2 = 0.900512 | 7.0E-06 | CAD | ZNF383 | 23870195 | 19, 37808501, 37809067 | HKR1 | common |

to ERG-bound ETS motifs and that AP1 motif presence decayed with distance from the ETS motif (*Figure 3a*). Reciprocally, ETS motifs were most frequently observed proximal (within 50 base pairs) to JUN/JUNB co-bound AP1 motifs (*Figure 3b*). Both AP1 and ETS motifs were frequently observed near GATA2-bound GATA motifs; however, neither GATA nor SOX motifs were prominent in the vicinity of ETS or AP1-bound motifs (*Figure 3c*). These data support that AP1 and ETS factors collaborate to determine the active chromatin landscape in HAECs with GATA and SOX serving less active roles genome-wide. This observation is consistent with GATA and SOX motifs only having enrichment in EC-specific enhancers (*Figure 1—figure supplement 3*).

## Allele-specific binding to chromosomes lacking motif mutations supports collaborative binding between AP1 and ETS factors

One approach to study collaborative binding between TFs is to knock-down/out a TF of interest and observe a shift in binding or activity at the regulatory element. To avoid complications in interpretability caused by potential redundancy of TF members, we took an alternative approach. As applied in inbred mouse strains previously (*Heinz et al., 2013*; *Gosselin et al., 2014*), we utilized naturally occurring genetic variation as a genome-wide source of motif mutations. The hypothesis is if the motif for a collaborative transcription factor is mutated then it should affect binding of the collaborating transcription factor whose motif remains in tact. To test this, whole-genome sequencing (WGS) of one HAEC donor was performed at an average of 40X coverage and the identified SNPs were phased with the appropriate 1000 Genomes (*Auton et al., 2015*) reference population (see Materials and methods). To quantify JUN binding to distinct homologous chromosomes at heterozygous loci, JUN ChIP-seq reads were iteratively mapped to human genome builds containing the appropriate allele. Sequence tags with discrepant mappings were omitted to avoid bias. For all loci with a JUN peak containing at least one heterozygous SNP, ChIP-seq tags were counted that could be uniquely assigned to one homologous chromosome. These data were then analyzed with respect to loci where only one SNP allele mutated either the AP1 or ETS motif.

Results showed that JUN binding was significantly affected by mutations in the AP1 motif ($p = 5.0e^{-10}$) such that binding was predominant on the chromosome lacking AP1 motif mutations and diminished on chromosomes containing the mutation (*Figure 3—figure supplement 1*). Interestingly, JUN binding was also significantly affected by mutations in the ETS motif that occurred within 100 base pairs of the JUN peak center (*Figure 3d p*-value = $2.6e^{-3}$). We would expect to observe the reciprocal relationship, in which AP1 motif mutations alter ERG binding, but the ERG ChIP-seq experiment in the sequenced HAEC donor yielded less than ten thousand peaks and more information is necessary for this analysis. Taken together, these data support a collaborative relationship between AP1 and ETS factors at endothelial enhancers.

## JUN and ERG co-occupy multiple elements near endothelial-specific genes

To interrogate the gene targets of JUN and ERG, we began with loci for genes expressed specifically or predominantly in ECs. All the genes queried, including vascular endothelial cadherin, (CDH5 or VE-cadherin), epidermal growth factor-like protein 7 (EGFL7), von Willebrand Factor (VWF),

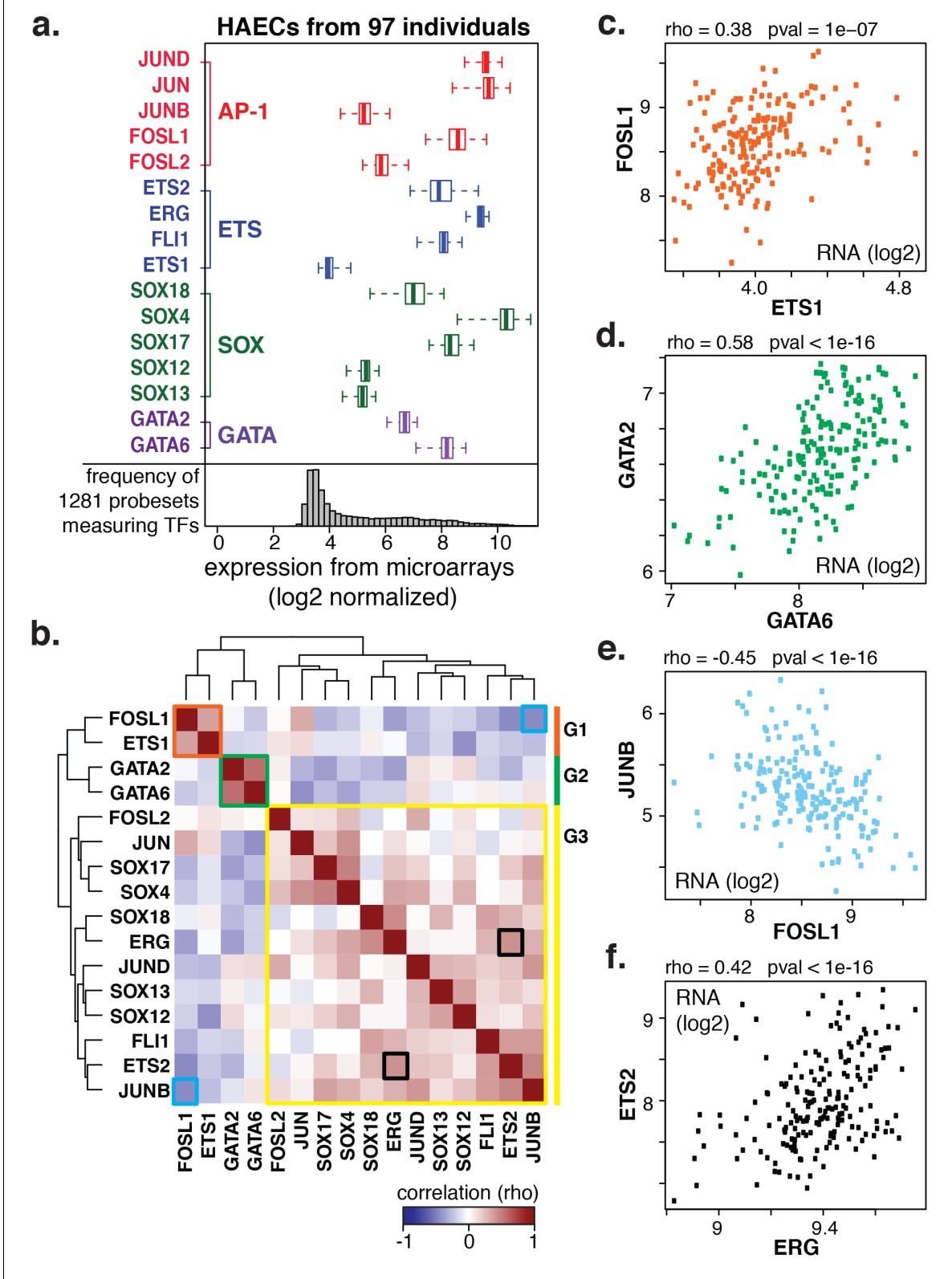

**Figure 2.** Coordinate gene expression across 96 genetically distinct HAEC donors identifies three regulatory programs among ETS, AP1, SOX, and GATA family members. (a) TF gene expression measured by AffyHU133A array is shown across a population of 97 unique HAEC donors. Array probe set IDs were manually confirmed to cover expressed transcript isoforms of the indicated TFs based on RNA-seq data. Boxplots midlines are medians, box edges are 1st and 3rd quartiles, and whiskers 95% confidence intervals. (b) A heatmap of clustered pairwise Spearman correlation coefficients

*Figure 2 continued on next page*

*Figure 2 continued*

across 97 HAEC donors and module designations (colored sidebar) is shown with modules and pairs of interest highlighted by colored outlines. (c–f) Pairwise Spearman correlations between indicated mRNAs from b where each dot is a genetically distinct HAEC donor. More related data in *Figure 2—figure supplements 1* and *2*.

The following figure supplements are available for figure 2:

**Figure supplement 1.** The HAEC gene correlation network.

**Figure supplement 2.** TF binding at HAEC enhancers.

endothelial nitric oxide synthase (NOS3), and TEK receptor tyrosine kinase (TEK, or TIE2), exhibited between three and seven ERG/JUN co-bound enhancers across their genetic loci (*Figure 3e*, *Figure 3—figure supplement 2*). By cross-referencing cistromes of ERG and JUN that individually bound 35,559 and 63,312 genomic loci respectively, we found that 10,919 of the 16,292 (65%) high confidence enhancers from *Figure 1a*. were bound by one or both of ERG and JUN (*Figure 3f*). Each enhancer was assigned a target gene(s) based on the following criteria. Since nearest genes are not necessarily the target of enhancer activity, we incorporated expression quantitative trait loci, or eQTL, that were identified in HAECs (*Romanoski et al., 2010*). eQTL are SNP-gene pairs that describe a genetic locus whose alleles are associated with quantitative levels of gene expression values of the target gene, and thus provide a functional link between DNA sequence and gene regulation. Only 7% of enhancers harbored an eQTL SNP, in which case the associated gene was considered the target. In the remaining cases, the nearest gene was used. Pathway analysis for the resulting 4396 target genes for the 4248 ERG and JUN co-bound loci revealed significant enrichment in 'Cardiovascular system development and function' (p = 6.1e$^{-37}$, *Figure 3f*). Together, these data support that ERG and JUN are major TFs at EC enhancers, and that their collaborative binding regulates expression of endothelial-specific genes important in vascular development and function.

## ERG knockdown elicits a pro-inflammatory gene expression profile in HAECs

To test the functional importance of ERG on target gene expression, we knocked-down ERG using siRNA in HAECs and measured gene expression changes with RNA-seq and RT-qPCR (*Figure 4*, *Figure 4—figure supplements 1* and *2*). ERG RNA was reduced to less than 40% of normal levels in three independent experiments and resulted in differential expression of up to 1000 transcripts (>4-fold, FDR < 5%) by RNA-seq. Functional enrichment analysis demonstrated that ERG target genes are significantly annotated for 'cell movement', 'breast or ovarian cancer', 'angiogenesis', 'development of vasculature', 'leukocyte migration', and other pro-inflammatory functions (p-values from 1e$^{-4}$ to 1e$^{-17}$, *Figure 4a,b*).

Among the most up-regulated genes caused by ERG knockdown were cytokines interleukin one alpha (IL1$\alpha$; 16-fold), interleukin one beta (IL1$\beta$; 68-fold), leukemia inhibitory factor (LIF; 13-fold), interleukin 6 (IL6; fourfold), granulocyte colony stimulating factor (CSF3; 41-fold), transforming growth factor beta 2 (TGF$\beta$2; fivefold) and other pro-inflammatory molecules including tissue factor (F3, 8-fold) (*Figure 4b,c*, *Figure 4—figure supplements 1* and *2*). In addition, EC-enriched genes that had multiple elements bound by ERG, such as CDH5, VWF, PECAM1, EGFL7, NOS3, and TEK were down-regulated upon ERG knockdown. To ensure that the inflammatory gene profile elicited by ERG knock-down was not a consequence of transfection itself or off-target effects, the profile resulting from six individual siERG oligos was measured along with two non-targeting scrambled siRNA controls and non-transfection controls (*Figure 4—figure supplement 2*). These data were reproducible and consistent with ablated ERG expression as the cause of pro-inflammatory expression profiles. Together, these data suggest that ERG normally functions to maintain EC-specific gene functions such as development and proliferation while at the same time suppressing inflammatory gene expression.

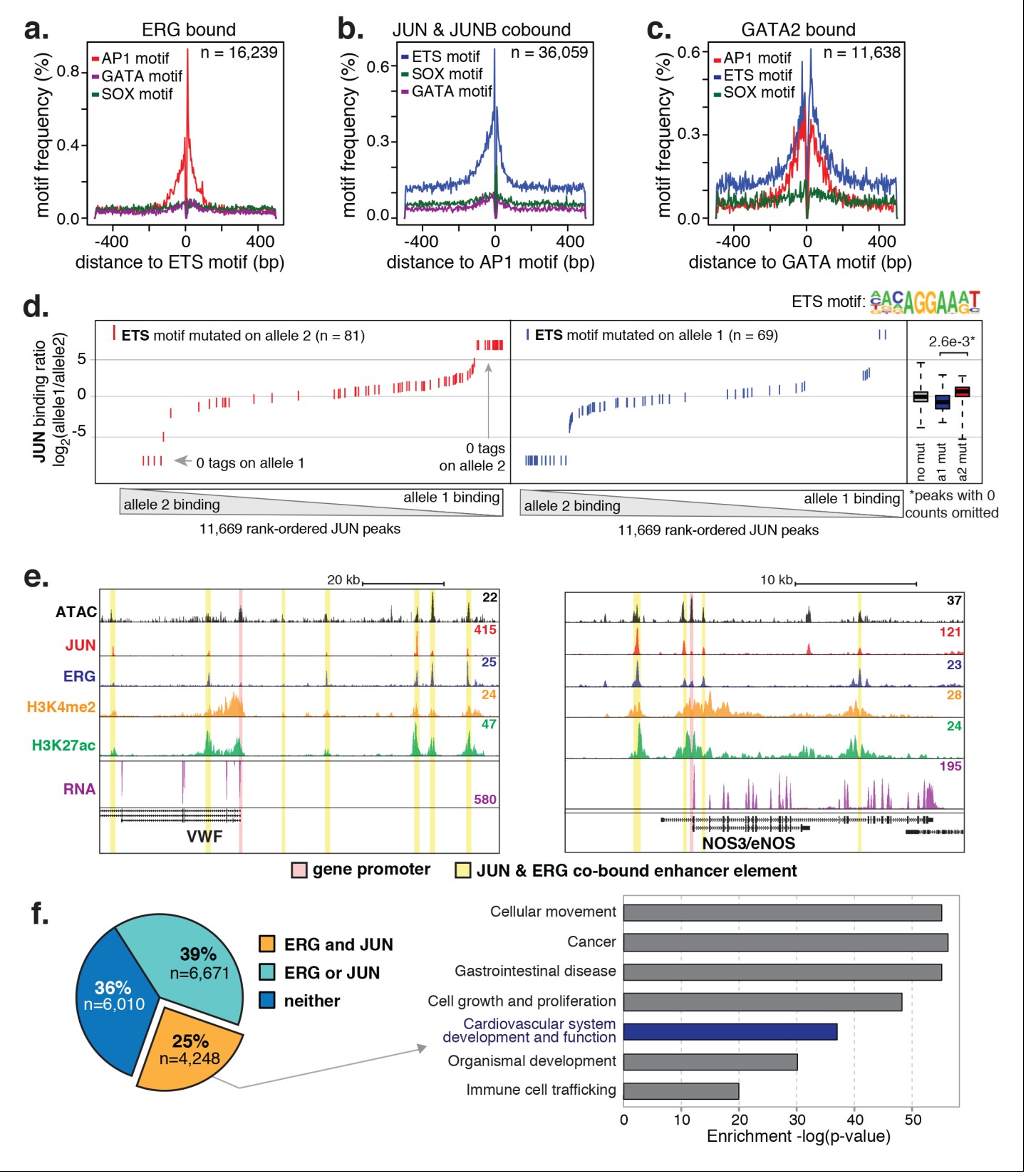

**Figure 3.** ERG and JUN co-bind EC enhancers and are enriched at EC-specific genes. (**a–c**) Promoter-distal regions bound by ERG, JUN, and JUNB, or GATA2 are shown in a, b, and c respectively in a one kilobase window. Each set was centered on the corresponding binding motif, and the frequency

*Figure 3 continued on next page*

*Figure 3 continued*

of other enriched motifs are shown on the y-axis. GATA2 binding was measured in HUVECs. (d) Allele-specific JUN binding (y-axis) as a function of allele-specific ETS motif mutations (colored lines). Each vertical line represents a single JUN peak identified via ChIP-seq. For a more complete explaination see the *Figure 3—figure supplement 1* legend and methods section. (e) TF binding for JUN and ERG, the histone modifications H3K4me2 and H3K27ac and RNA abundance are shown at the genetic loci for VWF and NOS3. Promoters are highlighted in pink and JUN/ERG co-bound enhancers are highlighted in yellow. (f) The HAEC enhancer set from *Figure 1a* is annotated for JUN and/or ERG binding. Ingenuity Pathway Analysis results are shown for the genes nearest the 4248 JUN/ERG co-bound enhancers in the right panel. More related data in *Figure 3—figure supplements 1* and *2*.

The following figure supplements are available for figure 3:

**Figure supplement 1.** Allele-specific JUN binding at loci with mutated motifs.

**Figure supplement 2.** JUN and ERG co-occupy loci near EC-specific genes.

## Oxidized phospholipids and inflammatory cytokines alter HAEC gene expression through signal-dependent changes to HAEC enhancer landscapes

To identify regulatory factors that instruct gene expression responses to inflammatory environments of atherosclerosis, we exposed HAECs to three pro-inflammatory treatments: (i) the oxidized products of 1-palmitoyl-2-arachidonoyl-sn-glycero-3-phosphocholine (oxPAPC) that are components of oxidized low-density lipoproteins (oxLDL) (*Lee et al., 2012*; *Romanoski et al., 2011*), (ii) tumor necrosis factor alpha (TNFα) that is a cytokine secreted largely by macrophages, and (iii) interleukin one beta (IL1$\beta$) that is released by many cell types including macrophages. Between 322 and 1174 genes were regulated by these exposures (>2-fold; 5% false discovery rate) (*Table 2*), and these genes were enriched in known response pathways (*Figure 5—figure supplement 1*). For example, 4 hr exposure to 40 µg/ml oxPAPC treatment resulted in up-regulation of genes belonging to the 'NRF2-mediated oxidative stress response' (p-value=2.e$^{-13}$), and the 'unfolded protein response' (p-value=7.0e$^{-7}$). Genes regulated by 4 hr exposures to TNFα and IL1$\beta$ were highly enriched in the same pathways including, 'TNF receptor signaling' (p-values from 1.7e$^{-14}$ to 1.8e$^{-12}$), 'granulocyte adhesion and diapedesis' (p-values from 2.2e$^{-9}$ to 7.1e$^{-13}$), and 'macrophage, fibroblast and EC roles in rheumatoid arthritis' (p-values from 2.1e$^{-12}$ to 1.7e$^{-13}$).

To better understand the program that coordinates HAEC response to oxPAPC, TNFα, and IL1$\beta$, we measured enhancer elements genome-wide. We next defined de novo or latent, enhancer-like elements that are genomic regions that became accessible and gained H3K27ac modification upon treatment. De novo enhancers result when signal-dependent transcription factors (SDTFs) play a critical role in the enhancer activation process and, in this study, were used in this study to identify SDTFs. We identified between 266 and 3199 de novo enhancers across treatments (*Figure 5a*, *Table 2*). To identity the SDTFs, we performed motif enrichment that revealed differential enrichment of TF motifs across treatments (*Figure 5b*; comprehensive list in *Figure 5—source data 1*). For example, the C/EBP, NFκB, and IRF motifs were preferentially enriched in TNFα and IL1$\beta$ enhancers, whereas the anti-oxidant response element, or ARE, was enriched in the oxPAPC enhancer set. In all sets, we found AP1 and ETS motifs were highly enriched (p<1e$^{-6}$), consistent with the model that the predominant AP1 and ETS endothelial factors collaborate with newly activated SDTFs to activate responsive enhancers and direct dynamic gene expression (*Figure 5a,b*). Enrichment of κB motifs at TNFα and IL1$\beta$ enhancers is consistent with previous work demonstrating that NFκB is a master transcription factor of inflammatory gene programs in s as well as other cell types (*Brown et al., 2014*). Likewise, oxPAPC-induced enrichment of the ARE motif, to which the TF nuclear factor, erythroid 2-like 2, or NFE2L2/NRF2 binds, is consistent with reports of single gene targets that the transcription factor NRF2 regulates the response to oxidative stress (*Ma, 2013*; *Jyrkkänen et al., 2008*).

To directly test the hypothesis that NRF2 and NFκB were in fact SDTFs responsible for inflammatory gene responses, we measured the NRF2 cistrome upon oxPAPC treatment and the NFκB cistrome (using ChIP-seq with κB-component p65) in TNFα and IL1$\beta$ 4-hr-treated ECs. NRF2 and p65 binding were associated with increases in H3K27ac on adjacent nucleosomes consistent with a role

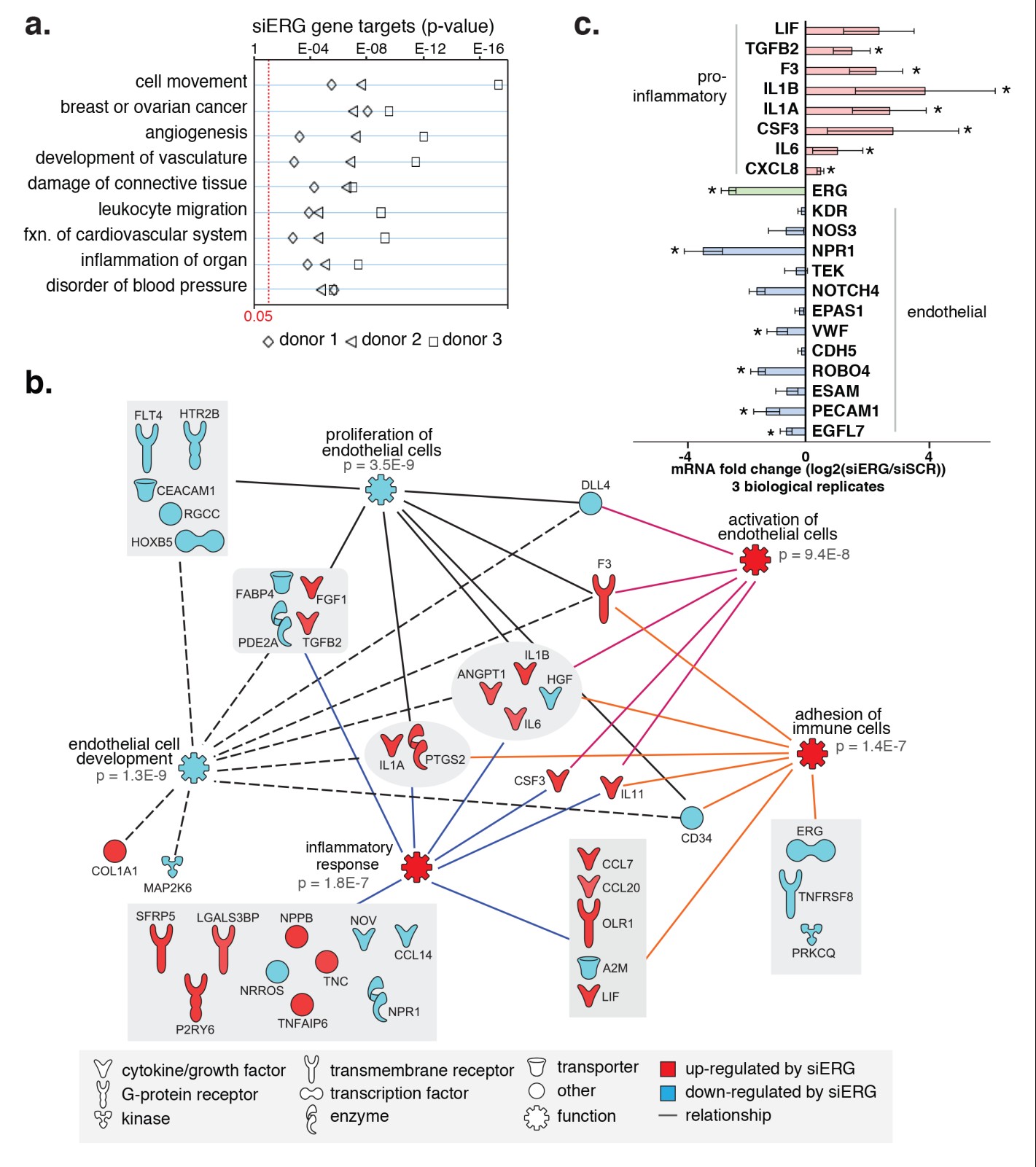

**Figure 4.** ERG knockdown elicits a pro-inflammatory gene profile in HAECs. (a) Genes up-regulated by ERG knockdown (≥4 fold, 5% FDR) were enriched in the indicated functional pathways using Ingenuity Pathway Analysis (IPA). Results are from three experiments using cells from different HAEC donors. ERG levels are shown in c. (b) A network of molecules regulated by ERG knockdown (red = up-regulated; blue = down-regulated) with membership (lines) to five functional processes. The predicted state of the functional processes is shown by color (activation in red; and inhibited in

*Figure 4 continued on next page*

*Figure 4 continued*

blue). Network analysis was performed using IPA and p-values correspond to pathway enrichments. (c) The log2 fold change caused by siERG knockdown is shown for endothelial-specific genes (blue) and pro-inflammatory genes (pink). Plotted is mean ± SD of fold changes measured in three HAEC donors. Per donor/gene effects are included in *Figure 5—figure supplement 1*. Asterisk indicates p-value<0.05 from a paired t-test of siERG to siSCR values across three donors. More related data in *Figure 4—figure supplements 1* and *2*.

The following figure supplements are available for figure 4:

**Figure supplement 1.** Changes in gene expression with ERG knockdown measured by RNA-seq.

**Figure supplement 2.** Changes in gene expression with ERG knockdown measured by RT qPCR.

for NRF2 and NFκB as activators of target gene expression (*Figure 5—figure supplements 2* and *3*). Genome-wide binding of NRF2 with oxPAPC treatment identified 6923 NRF2-bound peaks that overlapped with 21% of oxPAPC-elicited de novo enhancers. These included gene loci for known targets including heme oxygenase 1 (HMOX1), thioredoxin reductase 1 (TXNRD1), NAD(P)H quinone dehydrogenase 1 (NQO1), and glutamate-cysteine ligase modifier subunit (GCLM) (*Figure 5—figure supplement 2*). The top four enriched motifs in the NRF2 cistrome were the AP1 motif, an ARE, a nuclear receptor element (NRE) and the ETS motif. These data suggest that, whereas NRF2 is a significant component of the oxPAPC gene response, several additional TFs likely coordinate gene responses at the chromatin level.

For NFκB, we identified tens of thousands of bound loci after TNFα and IL1β treatment (75,937 and 51,040, respectively). Upon cytokine treatment, 52–63% of the elements that gained p65 were pre-bound in untreated HAECs by ERG or JUN (*Figure 5c*), consistent with the working model that enhancers are selected by lineage-restricted combinations of factors that direct signal-dependent transcription factor binding profiles (*Romanoski et al., 2015*). As for de novo enhancers, NFκB had a major presence and bound to 86% and 55% of those elicited by TNFα and IL1β treatments, respectively. Motif enrichment of NFκB cistromes in HAECs identified a prominent role of the AP1 motif as well as roles for ETS, IRF, and CEBP factors (*Figure 5—figure supplement 3*, discussed below).

In addition, we analyzed allele-specific NFκB binding as a function of motif mutations in the κB motif itself as well as mutations in the AP1 and ETS motif. Interestingly, we observed that NFκB binding was diminished at loci where κB, ETS, or AP1 motifs were mutated (*Figure 5—figure supplement 4*). These data are consistent with collaborative interactions between ETS, AP1, and NFκB in establishing inflammatory expression profiles in human ECs, and offer a mechanism whereby ECs may generate cell-specific transcriptional responses to environmental stimuli. Furthermore, these data demonstrate that allele-specific binding in heterogeneous human cells is a useful means to reveal collaborative interactions between transcription factors.

## Non-random spacing of ETS and κb motifs at co-bound elements suggest interplay between ERG and NFκB in inflammation

The transcriptional signature resulting from ERG knockdown in aortic ECs (*Figure 4*) is evidence that ERG performs an anti-inflammatory role. This has been demonstrated previously, where ERG suppressed IL-8 and NFκB-mediated inflammation in HUVECs (*Dryden et al., 2012*; *Sperone et al.,*

**Table 2.** Molecular trait changes observed upon HAEC exposure to pro-inflammatory stimuli. Differential expression was determined in DESeq with duplicate RNA-seq experiments.

| Stimulus | Regulated genes (>2 fold, 5%FDR) | de novo enhancers increased H3K27ac and accessibility upon stimulation |
|---|---|---|
| oxPAPC 4 hr (40 μg/ml) versus control | 322 (242 up, 80 down) | 839 |
| TNFα 4 hr (2 ng/ml) versus control | 840 (611 up, 229 down) | 266 |
| IL1β 4 hr (10 ng/ml) versus control | 1174 (807 up, 367 down) | 3199 |

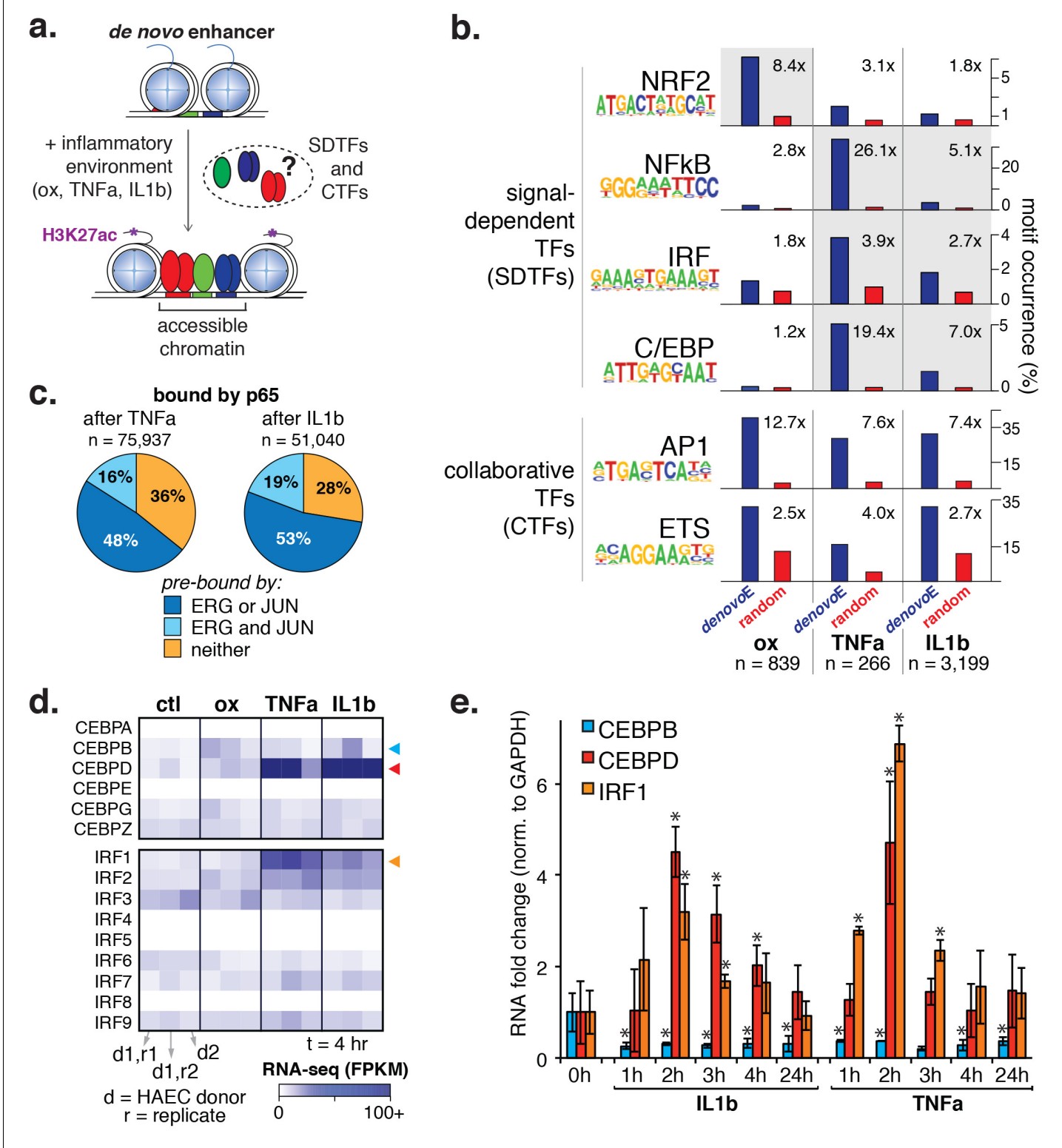

**Figure 5.** Inflammatory signals activate enhancer-like elements with distinct motifs and suggest that CEBP and IRF members mediate responsiveness to TNFα and IL1β. (a) A schematic for de novo enhancer activation by combinations of signal-dependent and collaborative TFs. (b) Enrichments of the NRF2, NFκB, IRF, C/EBP, AP1, and ETS motifs (y-axis) are shown for de novo enhancer sets activated after 4 hr by either 40 μg/ml oxPAPC (ox), 10 ng/ml TNFα, or 2 ng/ml IL1β (x-axis). Blue bars show the percent of de novo enhancers containing the motif and red bars indicate the percentage of GC-matched random genome sequence containing the motif. Motifs were considered within 200 basepair windows of enhancers. (c) ChIP-seq analysis of

*Figure 5 continued on next page*

*Figure 5 continued*

p65-binding sites shows regions of the genome that are co-bound by p65 and JUN, p65 and ERG, all three, or solely p65 post treatment by 10 ng/ml IL1$\beta$ and 2 ng/ml TNF$\alpha$ treatment. (**d**) Gene expression, measured by RNA-seq is shown by heatmap for TFs of the C/EBP and IRF families. Expression values are shown from two HAEC donors and replicate samples. (**e**) RT-qPCR analysis shows CEBPB, CEBPD, and IRF1 expression from 0 to 24 hr post treatment by 10 ng/ml IL1$\beta$ and 2 ng/ml TNF$\alpha$ treatment. Plotted are mean ± S.D., experiment performed with biological triplicates. * represents p<0.05 by t-test. More related data in *Figure 5—figure supplements 1–4*.

The following source data and figure supplements are available for figure 5:

**Source data 1.** Motif enrichments in 100 bp sequences defined by promoter-distal (≥3 kb) loci gaining ATAC-seq and H3K27ac upon 4 hr oxPAPC, TNF$\alpha$, and IL1$\beta$ treatments.
**Figure supplement 1.** The transcriptional response to oxPAPC, TNF$\alpha$, and IL1$\beta$.
**Figure supplement 2.** Signal-responsive transcription factor NRF2 binds endothelial enhancers.
**Figure supplement 3.** Signal-responsive transcription factor NF$\kappa$B binds endothelial enhancers.
**Figure supplement 4.** Binding of p65 at mutated motifs at each allele p65 (NF$\kappa$B) binding was measured by ChIP-seq in a HAEC donor with whole-genome sequencing information.

*2011*; *Yuan et al., 2009*). One model to explain the relationship between anti-inflammatory effects of ERG and the pro-inflammatory effects of NF$\kappa$B at co-bound loci involves the observation that ERG levels are decreased upon inflammatory stimuli ([*Yuan et al., 2009*] *Figure 6—figure supplement 1a*). Depletion of ERG, simultaneous with increased NF$\kappa$B concentrations, could result in a functional switch caused by stoichiometric competition between factors. At the pro-inflammatory target gene ICAM-1, Sperone et. al. demonstrated putative ETS-binding sites within the NF$\kappa$B motif at the gene promoter(*Sperone et al., 2011*). We also observe co-occupancy of ERG and NF$\kappa$B (p65) at the ICAM-1 promoter (*Figure 6—figure supplement 1b*).

To examine the genome-wide relationship between ERG and NF$\kappa$B occupancy upon inflammatory signaling, we compared binding profiles. Twenty-nine percent of ERG binding sites in untreated ECs gained p65 binding upon 4-hr treatment with TNF$\alpha$, and conversely, 25% of TNF$\alpha$-elicited p65 binding occurred at loci pre-bound by ERG (*Figure 6—figure supplement 1c*). All co-bound loci were centered on the presumed ETS motif to which ERG binds and the distribution of NF$\kappa$B motifs within 100 base pairs was calculated. This analysis revealed a distance relationship between ETS and NF$\kappa$B motifs that is consistent with ERG and NF$\kappa$B affecting each other's binding and activity to promote pro-inflammatory gene expression (*Figure 6—figure supplement 1d*). Importantly, however, we do not observe reduction in ERG binding after TNF$\alpha$ treatment, as would be expected if the factors were competing for the same element. This is exemplified at the ICAM-1 promoter and in the variability of TNF$\alpha$-induced changes to ERG and NF$\kappa$B binding genome-wide (*Figure 6—figure supplement 1e*). Together, these data support coordinated regulation of inflammatory pathways by ERG and NF$\kappa$B that likely involves multiple mechanisms.

## A role for CEBPD and IRF1 in the HAEC response to inflammatory cytokines

The result that CEBP and IRF motifs were preferentially enriched in TNF$\alpha$ and IL1$\beta$-induced enhancers suggested that TFs in these families direct the EC response to cytokines (*Figure 5b*). To identify the likely members, we examined relative expression levels with and without cytokine exposure and found expression of CEBP and IRF factors to be relatively low in untreated HAECs. Upon TNF$\alpha$ and IL1$\beta$ exposure, however, CEBPD and IRF1 transcription were highly induced (*Figure 5d*). In order for CEBPD and IRF1 to participate in enhancer activation, we reasoned they would need to be expressed prior to 4 hr when de novo motifs were measured. Indeed, a time-course experiment confirmed that both CEBPD and IRF1 RNAs were induced after cytokine treatment with a peak expression at 2 hr (*Figure 5e*).

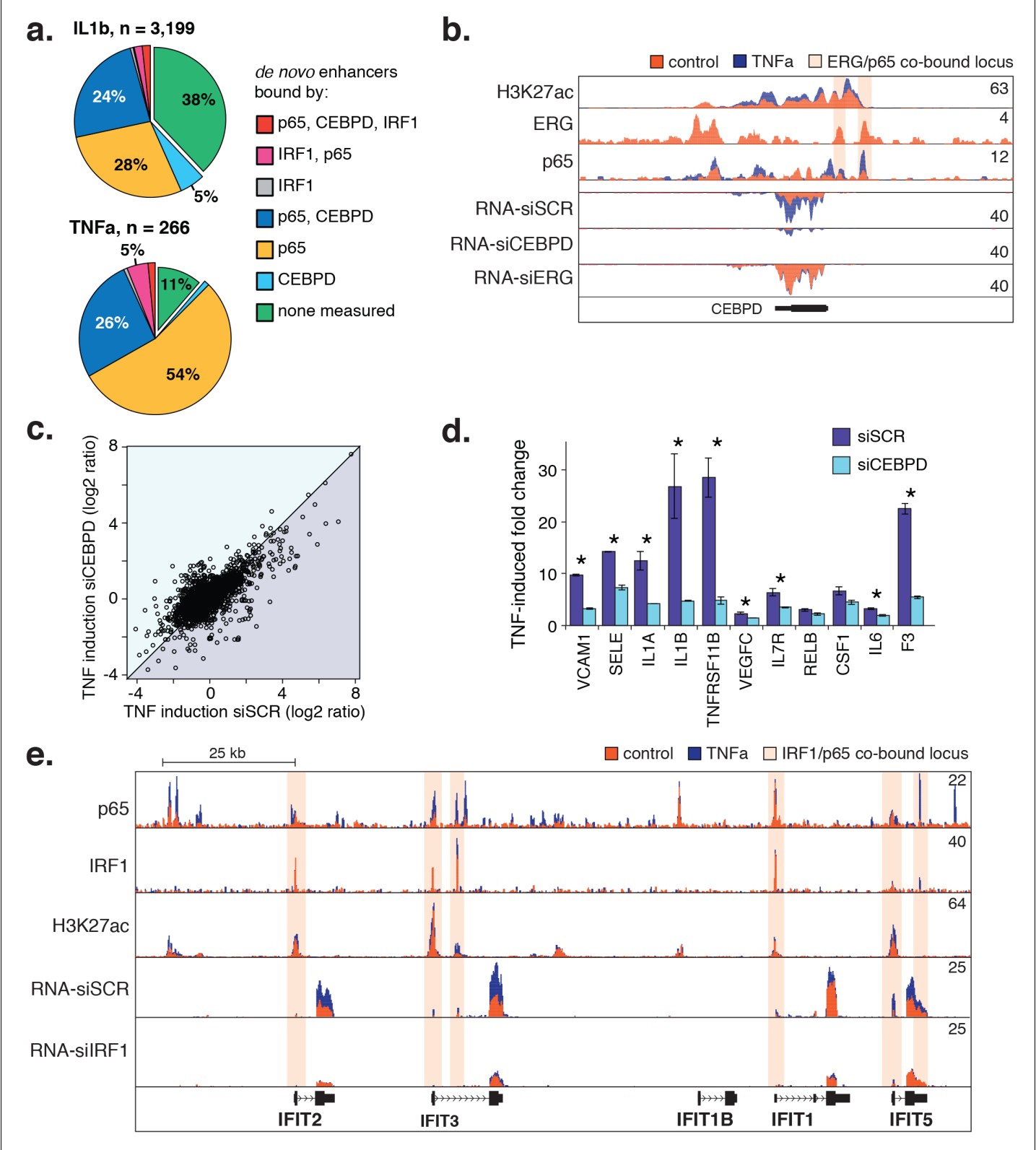

**Figure 6.** CEBPD and IRF1 knockdown effect on gene expression response to TNFα. (a) Binding of p65, CEBPD, and IRF1 was measured by ChIP-seq and is shown at de novo enhancers elicited by IL1β (e) and TNFα (f). Factor binding was measured by CHIP-seq upon 4-hr cytokine treatment, with binding of CEBPD measured after IL1β only. (b) Factor binding, H3K27ac, and mRNA expression is shown at the CEBPD locus in control-treated and TNFα-treated HAECs. CEBPD mRNA is shown in the bottom three tracks as a function of control knockdown (siSCR), CEBPD knockdown, and ERG

*Figure 6 continued on next page*

*Figure 6 continued*

knockdown. (c) A global view of mRNA responses to TNFα as a function of CEBPD knockdown (y-axis) compared to control (x-axis); mean ± S.D. (d) The response to TNFα is shown for molecules of interest in control (siSCR) and CEBPD knockdown HAECs. (e) The IFIT locus, highlighting elements co-bound by IRF1 and NFκB (p65). More related data in *Figure 6—figure supplement 1*.

The following source data and figure supplement are available for figure 6:

**Source data 1.** Transcripts up-regulated by more than twofold by CEBPD knockdown in untreated HAECs compared to scrambled control.
**Figure supplement 1.** ERG and p65 co-bind loci with a distinctive ETS/NFκB motif.

Next, we measured binding of IRF1 and CEBPD genome-wide after cytokine treatment (*Figure 6a*). For IRF1, we observed binding to the minority (<10%) of de novo enhancers, and when we did observe binding it was most frequently co-bound with either NFκB or CEBPD. On the other hand, CEBPD bound to a significant proportion of IL1$\beta$ and TNFα de novo enhancers (34% and 28%, respectively), mostly in conjunction with NFκB. Of all de novo enhancers, 24–26% were co-bound by NFκB and CEBPD within 200 basepairs of each other, strongly suggesting that these factors together regulate target gene expression.

Notably, ERG and NFκB bind to elements at the CEBPD locus and ERG knockdown induces CEBPD expression (*Figure 6b*). These data are consistent with a model whereby cytokine treatment causes down-regulation of ERG as well as nuclear entry and binding of NFκB. These two events induce CEBPD expression and enable CEBPD and NFκB to co-bind enhancers of target genes.

Lastly, we tested whether reducing CEBPD and IRF1 levels with siRNA would dampen the endothelial response to cytokines. As hypothesized, CEBPD knockdown to less than 20% control RNA levels resulted in dampened fold-change ratios for genes up-regulated by TNFα (250 genes up-regulated in siCEBPD compared to 497 in scrambled control; *Figure 6c*, bottom triangle). Genes in this set contained inflammatory molecules including vascular cell adhesion molecule 1 (VCAM1), E-selectin (SELE), IL1A, IL1$\beta$, and F3 (*Figure 6d*). However, lesser induction was partly due to an increase in these molecules in the untreated state (*Figure 6—source data 1*) suggesting that CEBPD may play an anti-inflammatory role at baseline. In the IRF1 knockdown, the most prominent locus affected was a stretch of related antiviral proteins called 'interferon-induced proteins with tetratrico-peptide repeats' (IFITs) on chromosome 10q23 (*Figure 6e*). We found that IRF1 bound six elements along this locus including at the promoters of IFIT2, IFIT3, IFIT1, and IFIT5. IRF1 knockdown reduced the basal levels of RNA from these genes and likewise prevented induction upon TNFα treatment, suggesting that IRF1 plays a critical role in their regulation. Together, these data provide evidence that CEBPD and IRF1 tune cytokine-induced regulatory function that is dominated by NFκB. However, further studies will be required to understand the precise role that IRF1 and CEBPD have in aortic EC gene regulation.

## Discussion

In this study, we provide the first detailed characterization of human aortic endothelial enhancers at baseline and under inflammatory conditions using a combination of genome-wide approaches. We observe that ETS, AP1, GATA, and SOX motifs are enriched in active EC enhancers, and we provide evidence that ETS and AP1 factors bind the majority of active enhancers in aortic ECs. Allele-specific binding paired with allele-specific motif mutations provided further evidence of collaborative binding between AP1, ETS, and κB factors. Our work demonstrates that knockdown of the ETS factor ERG results in a pro-inflammatory expression profile and corresponding down-regulation of EC-enriched genes. Further, we identify several hundred de novo enhancers formed in response to pro-inflammatory molecules abundant in the atherosclerotic plaque: namely, oxidized phospholipids and the cytokines TNFα and IL1$\beta$. Motif enrichments paired with expression changes prioritized NFκB, CEBPD, and IRF1 as the responsible coordinators of cytokine response and NRF2 as a coordinator of response to oxidized phospholipids. Our work provides valuable context to the role of these transcription factors within the regulatory networks controlling the endothelium in physiologic and disease states.

Our approach was to learn the regulatory lexicon of ECs beginning with H3K4me2-positive, H3K27ac-positive, accessible DNA sequence. In doing so we identified a set of TF families, each of which possesses many highly expressed members and inter-correlated expression patterns across aortic ECs from 97 people (*Figure 2*). Of note, whereas SOX members (SOX4/17/18) were among the most abundant of all TFs in aortic ECs, the SOX motif was not as highly enriched at EC enhancers as AP1 and ETS motifs (*Figures 1b, c* and *3a*). An explanation for this could be that SOX factors are critical for the selection of key endothelial enhancers in lineage development, and that only some of these enhancers remain open and available for other factor binding. This would explain the moderate enrichment of SOX motifs and is consistent with the role of SOX17 and SOX 18 in angiogenesis and vascular integrity (reviewed by [*De Val and Black, 2009*]). However, we acknowledge that this theory does not explain why ECs persistently express such high levels of SOX mRNAs. Another explanation, consistent with the selective enrichment of SOX and GATA motifs only in EC-specific enhancers (*Figure 1—figure supplement 3*) is that these factors play localized but important roles on target genes.

The finding that the ETS and AP1 factors ERG and JUN together bind the majority of EC enhancers, and co-bind key EC loci, supports a prominent and collaborative role for these factors in selecting endothelial specific enhancers. Our analysis of motif mutations further exemplifies a co-dependence of these factors in enhancer binding. While we focus on ERG and JUN in particular, other combinatorial interactions between other ETS and AP1 members almost certainly play prominent roles in endothelial regulation. Still, our findings are consistent with previous reports that ERG is among several ETS transcription factors critical for endothelial lineage development (*De Val and Black, 2009*; *Nikolova-Krstevski et al., 2009*; *Shah et al., 2016*) and regulation of candidate endothelial genes such as CDH5, VWF, and eNOS (*Yuan et al., 2009*; *Birdsey et al., 2008*; *Laumonnier et al., 2000*). Knockdown of ERG in aortic ECs revealed two notable effects, namely a reduction in EC-specific gene expression, and production of a pro-inflammatory transcriptional profile. The first of these effects is in keeping with our results demonstrating that ERG and AP1 co-bind promoters and enhancer elements at key endothelial gene loci. This supports our model whereby ERG binds collaboratively with AP1 factors to drive a basal lineage-defining transcriptional network in ECs.

Functional interactions between ETS and AP1 family members have been previously described (*Bassuk and Leiden, 1995*), and cooperative binding of ETS factors and FOS/JUN occurs at adjacent ETS and AP1 motifs with variable spatial orientation (*Kim et al., 2006*). In the current study, we also observe a spike in AP1 motifs near ETS motifs consistent with precise spatial orientation at some loci (*Figure 3a,b*), although this motif relationship does not describe the majority of genome-wide ERG and JUN co-binding we observe.

The transcriptional signature resulting from ERG knockdown in aortic EC also further supports the anti-inflammatory role for ERG, which has been demonstrated in HUVECs (*Dryden et al., 2012*; *Sperone et al., 2011*; *Yuan et al., 2009*). Our observation of regularly spaced ETS and κB motifs at co-bound loci suggests interplay between these two TF families. However, we do not observe coordinated change in genome-wide binding for ERG and NFκB upon TNFα treatment, indicating that these factors act in ways other than competitors for binding (*Figure 6—figure supplement 1*). In addition, we demonstrate ERG to regulate expression of numerous transcription factors (*Supplementary file 2*) that likely regulate secondary transcriptional targets. Recent work in murine EC has also demonstrated a role for ERG in promoting vascular integrity through promotion of canonical Wnt-signaling, via stabilization of β-catenin (*Birdsey et al., 2015*). This highlights the potential that ERG influences endothelial function through multiple mechanisms. Our present work underscores a role for ERG in promoting endothelial homeostasis and in regulating the inflammatory response. It also broadens our perspective of its regulatory function and cooperation with other factors genome-wide.

Our finding that the CEBP motif was enriched at TNFα and IL1β-induced enhancers and that CEBPD transcript levels were highly induced after treatment supported its role in binding inflammatory enhancers (*Figure 5*). CEBPD knockdown in aortic ECs in the absence of TNFα or IL1β caused modest up-regulation of pro-inflammatory molecules including ICAM1, SELE, IL1α, and IL1β (*Figure 6*, *Figure 6—source data 1*), suggesting that CEBPD maintains an anti-inflammatory expression profile in resting ECs. This is consistent with an anti-inflammatory role of CEBPD in pancreatic beta cells in the setting of cytokine stimulation (*Moore et al., 2012*). However, upon TNFα and IL1β

stimulation in aortic ECs, CEBPD knockdown also dampened this inflammatory response as compared to untreated ECs suggesting that in fact CEBPD is required for full responsiveness to cytokines (*Figure 6*). This is consistent with previous reports showing CEBPD to be up-regulated in many inflammatory settings including atherosclerosis (*Takata et al., 2002*; *Ko et al., 2015*). Going forward, a key part of elucidating the role of SDTF such as CEBPD will be to quantify their loss-of-function effect on de novo enhancers that arise upon inflammatory stimuli. Given our hypothesis of hierarchical transcription factor binding, we would expect to observe loss of some of these enhancers following SDTF inhibition. Complicating this approach, however, is that inflammatory stimuli often induce expression of SDTFs. For example, NFκB binds the promoter of CEBPD (*Figure 6b*). Therefore, targeted mutagenesis strategies, such as the CRISPR-Cas9 system, will be required to fully eliminate SDTF function and interpret the data. Overall, these findings motivate further experiments to fully understand the regulatory function of CEBPD.

Cell type and context-specific enhancer mapping is a critical approach toward fully understanding regulatory disease mechanisms, as many disease loci reside in non-coding DNA sequence. Nonetheless, enhancers are a challenge to measure in pure human cell types relevant to many diseases. We provide a list of candidate SNPs with potential EC-specific non-coding regulatory function (*Table 1*). This is an important step toward fully understanding the etiology and molecular pathogenesis of CAD in humans. In conclusion, our study of dynamic endothelial enhancer elements is an advancement toward a systems-levels understanding of vascular inflammatory diseases.

## Materials and methods

### Cell culture

HAEC were isolated as described (*Navab et al., 1988*) from aortic trimmings of donor hearts at the University of California, Los Angeles (UCLA). All HAECs were de-identified and exempt from consideration as human subjects research by institutional regulatory boards at UC San Diego and The University of Arizona. Cells were grown in culture in M-199 (ThermoFisher Scientific, Waltham, MA, MT-10–060-CV) supplemented with 1.2% sodium pyruvate (ThermoFisher Scientific, Catalog# 11360070), 1% 100X Pen Strep Glutamine (ThermoFisher Scientific Cat# 10378016), 20% fetal bovine serum (FBS, GE Healthcare, Hyclone, Pittsburgh, PA), 1.6% Endothelial Cell Growth Serum (Corning, Corning, NY, Product #356006), 1.6% heparin, and 10 µL/50 mL Amphotericin B (ThermoFisher Scientific #15290018). Cells were grown to 90% confluence in either 10-cm or 15-cm plates, and used primarily at passages 6 to 10. Cells were then treated with M-199 containing 1% FBS (control) or additionally containing either 40 µg/mL Ox-PAPC, 2 ng/mL human recombinant TNFα, or 10 ng/mL human recombinant IL1$\beta$ (cytokines from R&D Systems, Minneapolis, MN).

### Small-interfering RNAs and qPCR Primers

Knockdown of ERG, CEBPD, and IRF1 were performed using 1 nM siRNA oligonucleotides in Opti-MEM (ThermoFisher Scientific) with Lipofectamine 2000 (ThermoFisher Scientific). Transfections were performed in serum-free media for 4 hr, then cells were grown in full growth media for 48 hr. All siRNAs and qPCR primers used in this study are listed in *Supplementary file 3*.

### RNA-seq

HAECs were resuspended in RNA Lysis Buffer and RNA was extracted from cells using the Quick-RNA Micro Prep kit from ZymoResearch (Irvine, CA, #R1051), including optional DNase I treatment. mRNA was selected through poly-A isolation using Oligo d(T)25 beads (New England BioLabs, Ipswich, MA, #S1419S). Selected RNA was fragmented, followed by single strand cDNA synthesis using a SuperScript III First-Strand Synthesis System (ThermoFisher Scientific # 18080051), followed by second strand synthesis using DNA Polymerase I (Qiagen/Enzymatics, Beverly, MA, #P7050L). dsDNA ends were repaired with T4 DNA Polymerase (Enzymatics #P7080L). Barcode adapters (BIOO Scientific NEXTflex, Austin, TX, #514104) were ligated onto the ends of sequences using T4 DNA Ligase (Enzymatics #L-6030-HC-L) and samples were treated with Uracil DNA Glycosylase (UDG) (Enzymatics #G5010L). Libraries were then amplified by PCR (Phusion Hot Start II, ThermoFisher Scientific, #F549L) and purified (Zymo #D5205) for high-throughput sequencing.

## Chromatin immunoprecipitation sequencing (ChIP-seq)

ChIP-seq was performed as previously described (*Gosselin et al., 2014*). Briefly, HAECs were fixed at room temperature with 1% paraformaldehyde in PBS for 10 min, and then quenched with glycine. ChIPs for p65, JUN, and JUNB were performed from chromatin cross-linked by 2 nM Disuccinimidyl Glutarate Crosslinker (DSG) (ProteoChem, Hurricane, UT, #c1104) in PBS for 30 min followed by 1% paraformaldehyde in PBS for 15 min, and then quenched with glycine. Between 2 and 10 million cells were used for each ChIP-seq. Cell lysates were sonicated using a BioRuptor Standard or BioRuptor Pico (Diagenode, Belgium), and then immunoprecipitated using antibodies bound to a 2:1 mixture of Protein A Dynabeads (Invitrogen #10002D) and Protein G Dynabeads (Invitrogen #10004D). Antibodies used included H3K4me2 (EMD Millipore, Billerica, MA, #07–030), H3K27ac (Active Motif, Carlsbad, CA , #39135), CEBPD (Santa Cruz Biotechnology, Dallas, TX, #sc-636X), IRF1 (Santa Cruz #sc-497x), p65 (Santa Cruz #sc-372X), NRF2 (Santa Cruz #sc-1694X), JUN (Santa Cruz #sc-13032X), ERG (Santa Cruz #sc-354X), and JUNB (Santa Cruz #sc-73). Following immunopreciptation, crosslinking was reversed and libraries were prepared using the same method described for RNA-seq beginning with dsDNA end repair and excluding UDG. For each sample condition, an input library was also created using an aliquot of sonicated cell lysate that had not undergone immunoprecipitation. These samples were sequenced as below and used to normalize ChIP-seq results.

## Transposase-accessible chromatin using sequencing (ATAC-seq)

ATAC-seq was performed on 50,000 HAEC nuclei according to the original published protocol (*Buenrostro et al., 2013*) with the exception of size selection (125–175 base pairs on TBE gel) prior to sequencing to enrich for enhancer elements.

## Sequencing data samples, mapping, and normalization

Libraries were sequenced on an Illumina HiSeq 4000 according to manufacturer's specifications at the University California San Diego and at the University of Chicago. Public data was downloaded from public repositories and processed exactly as new data in this study (see below). Reads from ChIP-seq and ATAC-seq were mapped to the hg19 build of the human genome with Bowtie2 (*Langmead and Salzberg, 2012*) and RNA-seq reads were mapped with STAR (*Dobin et al., 2013*). For ATAC-seq, reads mapping to mitochondrial DNA were discarded. Mapped reads were organized into HOMER's preferred data structure called Tag Directories using the 'makeTagDirectory' command.

ATAC-seq and RNA-seq experiments that measured accessibility and expression in different treatment stimuli (control, oxPAPC, IL1$\beta$, and TNF$\alpha$) were all conducted at least twice. ERG knockdown was performed in three different HAEC donors. CEBPD and IRF knockdowns were performed in one HAEC donor. The Benjamini-Hochberg false discovery rate (FDR) method was used to correct for all multiple testing in this study. No explicit power analysis was used to compute sample size. Instead, genome-wide features such as enhancer elements served as the replicates for motif analysis and duplicate ChIP-seq experiments confirmed that binding peaks were consistent. With the exception of CEBPD and JUNB that were performed one time, all ChIP-seq experiments were performed in biological duplicate, meaning separate cell expansion and collection.

## RNA-seq analysis

For quantification, RNA-seq was normalized using Reads Per Kilobase of transcript per Million mapped reads (RPKM) procedure in HOMER (*Heinz et al., 2010*). RNA sequencing tags were only considered when they mapped to the same DNA strand as indicated by RefSeq annotation. Further, only tags in exons of genes were incorporated as to remove bias created by variable intron sizes. Together, RNA quantification was achieved using the HOMER command 'analyzeRepeats rna – strand + -count exons –rpkm'. In this study, RPKM is synonymous with FPKM (Fragments per kilobase mapped). Statistically significant differential expression from RNA-seq experiments was determined first by unnormalized counts (with analyzeRepeats -noadj) followed by statistical testing in DESeq (*Anders and Huber, 2010*) and a restriction to a 5% False Discovery Rate. Pathway enrichment analysis was performed using Ingenuity Pathway Analysis software (Qiagen).

## Peak calling

ChIP-seq and ATAC peaks were identified using un-immunoprecipitated chromain, or 'input', as a negative control. Inputs from the corresponding crosslinking condition were used for each ChIP. No input was used to call ATAC peaks. Peaks were identified in HOMER with the findPeaks program according to the data type. Transcription factor peaks were called using the 'findPeaks -style factor –size 200' option and histone peaks called using the 'findPeaks -style histone' option. ATAC-seq peaks were called using findPeaks with '-L 8 F 8 -style histone -size 75 -minDist 75 -minTagThreshold 6' options. Differential peaks between experiments were determined using the 'getDifferentialPeaks program with default parameters' (normalized tag count difference >4 fold and poisson enrichment p-value<0.0001).

Peak merging was performed in HOMER using the 'mergePeaks' program. For enhancers defined in *Figure 1a*, the ATAC-seq peak file was merged with peak regions defined in H3K4me2 and H3K27ac ChIP-seq experiments using the option '-d given' that requires overlap of genomic coordinates between the three peak files. Because the ATAC set was listed first, enhancers were centered on the center of the accessible region. Distal peaks throughput the study were defined as being at least three kilobases from the transcriptional start site of a gene (RefSeq hg19 definitions) using HOMER's 'getDistalPeaks.pl' command.

## De novo enhancers and super-enhancers

De novo enhancers were defined as loci that gained ATAC-seq and H3K27ac ChIP-seq in the same cell simulation. Individual gained peak sets was determined by 'getDifferentialPeaks', explained above, and significant sets were intersected using 'mergePeaks' where the ATAC-seq gained peaks were listed first to maintain centering on accessibility.

Super-enhancers were defined in HOMER from H3K27ac ChIP-seq experiments and corresponding inputs using 'findPeaks -style super -L 0'. This procedure follows the same logic as the original definition proposed by the Young laboratory (*Whyte et al., 2013*). Briefly, the implementation in HOMER identifies 'typical' ChIP-seq peaks, stitches proximal enhancers together, ranks the resulting enhancers by normalized tag counts over input, and thresholds enhancers above a flex-point (slope >1) as super-enhancers.

## Motif enrichment and distance analysis

Motif enrichment analysis was performed on peak sets using HOMER's 'findMotifsGenome.pl' program. As a background control, this program selects a set of sequences from the same genome build that are matched in size and GC content to the peak set of interest. In *Figure 5b*, to enable enrichment comparison for motifs across multiple peak sets, we used HOMER's 'findMotifsGenome. pl –mknown <motifs>' option iteratively across each de novo enhancer set. For each motif analysis, the amount of sequence analyzed depended on the data type and is indicated in the text.

Analysis of motif distances in *Figure 3a–c* was performed in peak regions identified by ChIP-seq (e.g. ERG-bound in 3a; AP1 bound in 3b; GATA2-bound in 3c). Each peak set was centered on the highest-scoring TF motif that matched the factor immuno-precipitated, so that 0 bp on the x-axis was the beginning of the likely bound motif (e.g. the ETS motif for ERG-bound in 3a). Peaks lacking consensus motifs for the respective factors were excluded from this analysis. Next, the frequency of the other motifs queried (e.g. AP1, GATA and SOX motifs in 3a) were calculated in HOMER by 'annotatePeaks.pl –hist –m' options and the frequency of the additional motifs tested in the vicinity were plotted to show positional relationships among motifs and surrounding genomic sequence.

## Whole-genome sequencing, motif mutation analysis and allele-specific factor binding

For the HAEC line sequenced for analysis as in *Figure 3d*, genomic DNA was prepared for paired-end WGS on Illumina HiSeqX (Illumina, CA) by Novogene (Sacramento, CA) according to manufacturer specifications. More than 80 million raw sequencing reads and a raw depth of 41X were obtained. Data were mapped to the hg19 genome using Burrows-Wheeler Aligner (*Li and Durbin, 2009*). Single nucleotide variants (SNVs) were identified using GATK (*DePristo et al., 2011*). SNVs were then compared to 1000 Genomes reference populations (*Auton et al., 2015*) and the HAEC donor clustered with the Mexican American reference (MXL) population by multi-dimensional scaling

analysis in PLINK (*Purcell et al., 2007*). SNVs were then phased according to the 1000 Genomes MXL reference population in BEAGLE (*Browning and Browning, 2016*; *Browning and Browning, 2007*).

Allele-specific binding of JUN and NFκB/p65 was quantified using the WASP pipeline (*van de Geijn et al., 2015*). To avoid mapping bias caused by alternate alleles, reads that mapped discordantly to reference and alternate alleles at heterozygous SNPs were discarded. Next, sequencing reads from ChIP experiments were aligned to each haplotype at polymorphic loci and summed within ChIP-seq peaks for the respective factor.

Allele-specific motif mutations were identified with the same method as reported previously (*Heinz et al., 2013*; *Gosselin et al., 2014*) with slight modifications. In summary, reference genome sequence (hg19) was extracted at peaks of interest and intersected with SNVs in the HAEC donor of interest. Phased variant data was pulled for each homologous chromosome and alleles were inserted in turn to the genome sequence. Motifs were located by alignment to position weight matrices of ETS, AP1, and κB motifs in HOMER (*Heinz et al., 2010*). Motif mutations were defined as instances where motifs were only identified on one of the two homologous chromosomes.

For visualization, plots as in *Figure 3d* show the relationship between allele-specific TF binding and mutations in motifs of interest within the TF peaks containing at least one variant. TF binding was transformed to the log2 scale and peaks containing 0 reads on one of the two alleles were scaled to ±7.5 (or else the ratio would be ±Infinity). Boxplots showing the distribution of read ratios in each category (no mutations, mutation on allele 1, mutation on allele 2) exclude peaks with 0 counts on either chromosomal pair. This is a conservative approach, as many peaks have zero reads on the mutated chromosome.

## Data visualization

Heatmap-syle histograms of sequencing tags (e.g. *Figure 1a*), were generated using HOMER's 'annotatePeaks.pl -ghist' option and plotted in R using the 'heatmap.2()' function of the gplots library. Cumulative histograms of tag frequencies by position to peak center (e.g. *Figure 1c*), were generated using HOMER's 'annotatePeaks.pl –hist' option. Standard heatmaps (e.g. *Figure 2b*) were plotted in R using 'heatmap.2()' with default clustering parameters (Hierarchical clustering and Euclidian distance).

## Enhancer overlap with GWAS data

Coronary artery disease, hypertension, and related traits were downloaded from the NHGRI-EBI Catalog of published genome-wide association studies (*Welter et al., 2014*). SNPs in linkage disequilibrium (LD) to GWAS association traits were calculated when r2 >0.8 in PLINK (*Purcell et al., 2007*) according to the European reference population of the 1000 Genomes Project (*Auton et al., 2015*). HAEC enhancers defined in *Figure 1a* were overlapped by physical position (hg19 genome build). The studies reporting associations that overlapped EC enhancers are as follows: (*Nikpay et al., 2015*; *Samani et al., 2007*; *Coronary Artery Disease (C4D) Genetics Consortium, 2011*; *Schunkert et al., 2011*; *Wild et al., 2011*; *Slavin et al., 2011*; *Aouizerat et al., 2011*; *Mehta, 2011*; *Davies et al., 2012*; *Wojczynski et al., 2013*; *Dichgans et al., 2014*; *Kertai et al., 2015*).

## Public datasets

With the exception of data used to generate *Figure 1—figure supplement 3* (see *Supplementary file 1* for list), raw sequencing data was downloaded from Gene Expression Omnibus (GEO) as short read archive files and converted to fastq files using 'fastq-dump'. Fastq files were mapped to the hg19 genome build in Bowtie2 and processed according to methods outlined for corresponding data types above. The following publicly available datasets were analyzed: GSE52642 (HAEC GRO-seq), GSE20060 (HAEC microarrays), GSE41166 (ETS-1 HUVEC ChIP-seq), and GSE31477 (GATA2 HUVEC ChIP-seq).

Data generated in this study is available under GEO accession GSE89970 and WGS via NCBI SRA Archive, BioProject PRJNA381088.

## Cell lines

The HAECs used in this study were primary isolates of ECs from aortic trimmings collected from trimmings of donor hearts transplanted through the UCLA heart transplant program as previously described (*Navab et al., 1988*). No transformations were performed and they were used at low passage (p6-p10).

## Acknowledgements

CER was supported by NIH grant R00123485 and CKG by NIH grants DK091183, DK063491, and GM085764. NH was supported as a Research Fellow by the Sarnoff Cardiovascular Research Foundation. JRS was supported by NIH grant 1R15HL121770-01A1.

## Additional information

### Competing interests

CKG: Reviewing editor, *eLife*. The other authors declare that no competing interests exist.

### Funding

| Funder | Grant reference number | Author |
| --- | --- | --- |
| National Institutes of Health | R00123485 | Casey E Romanoski |
| Sarnoff Cardiovascular Research Foundation | Research Fellowship | Nicholas T Hogan |
| National Institutes of Health | 1R15HL121770-01A1 | James R Springstead |
| National Institutes of Health | DK091183 | Christopher K Glass |
| National Institutes of Health | DK063491 | Christopher K Glass |
| National Institutes of Health | GM085764 | Christopher K Glass |

The funders had no role in study design, data collection and interpretation, or the decision to submit the work for publication.

### Author contributions

NTH, Conceptualization, Investigation, Writing—original draft, Project administration, Writing—review and editing; MBW, Investigation, Writing—original draft, Writing—review and editing; LKS, Formal analysis, Investigation, Writing—original draft, Writing—review and editing; NKH, Formal analysis, Validation, Investigation; MTL, Conceptualization, Investigation; JRS, Resources, Methodology; CKG, Conceptualization, Resources, Writing—review and editing; CER, Conceptualization, Resources, Data curation, Formal analysis, Supervision, Funding acquisition, Investigation, Visualization, Methodology, Writing—original draft, Project administration, Writing—review and editing

### Author ORCIDs

Christopher K Glass, http://orcid.org/0000-0003-4344-3592
Casey E Romanoski, http://orcid.org/0000-0002-0149-225X

## Additional files

### Supplementary files

• Supplementary file 1. Public H3K27ac ChIP-seq datasets used to define endothelial-specific enhancer-like regions. The GEO accession number and cell or tissue type is listed for each dataset in the analysis.

• Supplementary file 2. Knockdown of ERG with siRNA modulates transcription of numerous other HAEC transcription factors. Above are the transcription factors shown to be modulated in two separate HAEC donors (by poly-A RNA-sequencing, greater than twofold change in transcript level, adjusted p<0.05).

• Supplementary file 3. List of primers and siRNA oligos used in study

## Major datasets

The following datasets were generated:

| Author(s) | Year | Dataset title | Dataset URL | Database, license, and accessibility information |
|---|---|---|---|---|
| Romanoski CE, Hogan NT | 2017 | Genome-wide map of HAEC chromatin landscape under resting and TNFa, IL1b, and OxPAPC stimulation, with corresponding transcription factor binding and RNA expression | https://www.ncbi.nlm.nih.gov/geo/query/acc.cgi?acc=GSE89970 | Publicly available at the NCBIGene Expression Omnibus (accession no: GSE89970) |
| Romanoski CE, Hogan NT | 2017 | Transcriptional networks specifying homeostatic and pro-inflammatory programs of gene expression in human aortic endothelial cells | http://www.ncbi.nlm.nih.gov/bioproject/381088 | Publicly available at the NCBIBioProject database (accession no: PRJNA381088) |

The following previously published datasets were used:

| Author(s) | Year | Dataset title | Dataset URL | Database, license, and accessibility information |
|---|---|---|---|---|
| Kaikkonen MU, Niskanen H, Romanoski CE | 2014 | Control of VEGF-A trancriptional programs by pausing and genomic compartmentalization | https://www.ncbi.nlm.nih.gov/geo/query/acc.cgi?acc=GSE52642 | Publicly available at the NCBIGene Expression Omnibus (accession no: GSE52642) |
| Romanoski CE, Lusis AJ | 2010 | Expression of human aortic endothelial cells treated with or without oxidized phospholipids | https://www.ncbi.nlm.nih.gov/geo/query/acc.cgi?acc=GSE20060 | Publicly available at the NCBIGene Expression Omnibus (accession no: GSE20060) |
| Zhang B, Day DS, Ho J, Song L, Cao J, Crawford GE, Park PJ, Pu WT | 2013 | A dynamic H3K27ac signature defines VEGF-regulated endothelial enhancers | https://www.ncbi.nlm.nih.gov/geo/query/acc.cgi?acc=GSE41166 | Publicly available at the NCBIGene Expression Omnibus (accession no: GSE41166) |
| Snyder M, Gerstein M, Weissman S, Farnham P, Struhl K | 2011 | ENCODE Transcription Factor Binding Sites by ChIP-seq from Stanford/Yale/USC/Harvard | https://www.ncbi.nlm.nih.gov/geo/query/acc.cgi?acc=GSE31477 | Publicly available at the NCBIGene Expression Omnibus (accession no: GSE31477) |

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
