## [Decision Letter]

Thank you for submitting your article "Transcriptional networks specifying homeostatic and inflammatory expression programs in human aortic endothelial cells" for consideration by *eLife*. Your article has been reviewed by three peer reviewers, one of whom is a member of our Board of Reviewing Editors, and the evaluation has been overseen by Harry Dietz as the Senior Editor. The following individual involved in review of your submission has agreed to reveal her identity: Lynn Butler (Reviewer #2).

The reviewers have discussed the reviews with one another and the Reviewing Editor has drafted this decision to help you prepare a revised submission.

Summary:

Hogan et al. performed an extensive analyses of transcriptional regulation in human aortic endothelial cells (HAECs). They authors provide a comprehensive map of the gene regulatory landscape of HAEC cells under basal (part1) and inflammatory conditions (part2). Using a combination of enhancer annotation, in-silico motif analysis and transcription factor ChIP the authors find that a Jun (an AP1 factor) and ERG (an ETS factor) cooperate to establish much of the cell enhancer landscape. In the second part, the authors study the transcriptional network in response to inflammatory stimulus.

Essential revisions:

1) It is unclear why the focus is only on JUNB. The reviewers felt there the authors should explain why JUND is not pursued further.

2) Two reviewers, one of them from prior experience, are worried that some of the observed effects are due to the transfection itself, or other off-target effects. The authors should carefully establish this before they conclude that ERG, or any other siRNA knock-down is the cause of the (observed) inflammatory response. The reviewers strongly suggest a combination of appropriate controls and replication with different siRNAs to ensure that the result is not due to secondary effects not related to the result of the ERG and all other knock-down experiments.

3) The collaborative nature of JUNB and ERG function would be strengthen by showing the effect of loss of function of JUNB.

[Editors' note: further revisions were requested prior to acceptance, as described below.]

Thank you for resubmitting your work entitled "Transcriptional networks specifying homeostatic and inflammatory expression programs in human aortic endothelial cells" for further consideration at *eLife*. Your revised article has been favorably evaluated by Harry Dietz (Senior editor) and a Reviewing editor.

The manuscript has been improved but there are some remaining issues that need to be addressed before acceptance, as outlined below:

There is one remaining point you may want to discuss. While the AP1 factors are highly expressed, the ETS factors are not. Figure 4—figure supplement 1 shows ERG expression to be less than 2.5 FPKM. Your data shows ERG and JUN co-binding on most regions. Is the protein of ERG that stable? Are the stoichiometry relationships reasonable? Clearly, at their mRNA levels ERG and JUN are at very different, is there evidence that ERG is a more stable protein? A related point: In Figure 3—figure supplement 2, Two of the three examples (ERK and TEK) have very small fold changes in siERG. What about other genes that show stronger effects?

[Editors' note: further revisions were requested prior to acceptance, as described below.]

Thank you for resubmitting your work entitled "Transcriptional networks specifying homeostatic and inflammatory expression programs in human aortic endothelial cells" for further consideration at *eLife*. Your revised article has been favorably evaluated by Harry Dietz (Senior editor), a Reviewing editor, and two reviewers.

The manuscript has been improved but there is one remaining issue that need to be addressed before acceptance, as outlined below:

While the methods for quantification of RNA-Seq data have been clearly improved since the original submission, there was consensus that these methods are not adequately described in the revised manuscript. Please expand and refine this description in the Materials and methods section.

---

## [Author Response]

*Essential revisions:*

*1) It is unclear why the focus is only on JUNB. The reviewers felt there the authors should explain why JUND is not pursued further.*

We agree that JUND is likely to be one of the important AP-1 family members involved in regulating endothelial transcription. This is evident from motif analysis of the 17 thousand HAEC enhancers that revealed a prominent role in AP1 transcription factors (Figure 1, Figure 1—figure supplement 1), high levels of JUND mRNA in HAECs (Figure 1, Figure 1—figure supplement 1) and the presence of a super-enhancer at the JUND locus (Figure 1—figure supplement 2, middle panel). We proceeded without JUND binding in the original manuscript because we had information about the binding of the other two AP1 factors (JUN and JUNB) and reasoned that with the redundancy in binding between factors and the heterodimeric binding property of AP1 factors that these two factors would be sufficient to provide considerable insight into functional roles of AP-1 as a whole. Consistent with this, we observe that the cistromes for JUN and JUNB are nearly identical (Figure 2—figure supplement 1). We did, however, fail to explicitly state our reasoning – an oversight that has now been corrected and is evident in the following sections of the revised manuscript:

Results section: “Tens of genes encode proteins of the AP1, ETS, SOX and GATA TF families. Within each family different members share nearly identical DNA binding domains and thus bind the same motif. In addition, AP1 protein members bind AP1 motifs as homo- and hetero-dimers. These sources of redundancy make it challenging to identify the functional family member(s) without additional information.”

Results section: “AP1 family members JUND, JUN, and JUNB were also in the top 4%. RNA-seq from other HAEC donors and replicate samples confirmed these findings (Figure 1—figure supplement 1). Next, we defined SEs using H3K27ac ChIP-seq data and found that, among others, the genetic loci for ERG (an ETS member) as well as AP1 members JUN, JUND, and JUNB harbored SEs (Figure 1, Figure 1—figure supplement 2). Taken together, these data suggest that while multiple TFs from each family probably bind HAEC enhancers, that JUN, JUNB, JUND and ERG likely serve prominent roles.”

Results section: “The JUND cistrome would also be informative in these studies; however, we proceeded with JUN and JUNB because the heterodimeric binding of AP1 factors makes it likely that JUN and JUNB profiles encompass a major portion of the overall AP1 landscape. Consistent with this, the JUN and JUNB cistromes are highly concordant (Figure 2—figure supplement 2).”

*2) Two reviewers, one of them from prior experience, are worried that some of the observed effects are due to the transfection itself, or other off-target effects. The authors should carefully establish this before they conclude that ERG, or any other siRNA knock-down is the cause of the (observed) inflammatory response. The reviewers strongly suggest a combination of appropriate controls and replication with different siRNAs to ensure that the result is not due to secondary effects not related to the result of the ERG and all other knock-down experiments.*

This point is well taken. Additional experiments were performed (refer to Figure 4—figure supplement 2) by each of the 4 individual siRNAs that comprised the pooled siERG mix used in the original submission (siERGs #1-4) as well as two additional single ERG-targeting siRNAs (siERG#5-6). In addition, two non-targeting scrambled oligos were used for comparison (siSCR#1-2) along with negative controls including Lipofectamine2000 only, oligo only, and optiMEM only, and full growth media changes only. qPCR for ERG was used to confirm knock-down and 1 of the 6 siERG oligos failed to reduce ERG message (siERG #6). In addition, inflammatory genes LIF, IL8, IL6, F3, IL1B, IL1A, CSF3, and CCL2 were measured across experimental conditions in three biological replicates. Data show that complete transfection complexes, e.g., siSCRs, slightly increased inflammatory messages for some genes. For example, IL8 levels were ~2.5 fold induced by siSCR compared to the non-transfected controls. This was in contrast to the robust up-regulation of inflammatory gene expression that was reproducibly observed across individual oligos targeting ERG. siERG6, which failed to reduce ERG levels, served as a control in that inflammatory expression was not induced for this condition. In addition, we measured 4 endothelial-specific genes in this experiment: VWF, PECAM1, EGFL7 AND NOS3. VWF was reproducibly significantly reduced upon ERG knock-down and levels of the other genes was also reduced – however, less dramatically than VWF. Our conclusion from these experiments is that ERG knock-down was the cause of the pro-inflammatory gene expression profile we (and others) have observed in endothelial cells. This information has now been added to the revised manuscript as follows:

Results section: “To ensure that the inflammatory gene profile elicited by ERG knock-down was not a consequence of transfection itself or off-target effects, the profile resulting from six individual siERG oligos was measured along with two non-targeting scrambled siRNA controls and non-transfection controls (Figure 4—figure supplement 2). These data were reproducible and consistent with ablated ERG expression as the cause of pro-inflammatory expression profiles. Together, these data suggest that ERG normally functions to maintain EC-specific gene functions such as development and proliferation while at the same time suppressing inflammatory gene expression.”

*3) The collaborative nature of JUNB and ERG function would be strengthen by showing the effect of loss of function of JUNB.*

We agree that an implication of collaborative TF binding is that loss-of-function of a participating TF would decrease the ability of the other factor to bind target elements. The expectation to observe this effect genome-wide is complicated, however, by other TFs in the same families that can bind the same DNA-binding motif and provide a surrogate collaborative role. Considering that we observe multiple AP1 and ETS members with high expression in HAECs, we decided to undertake an alternative approach to test for collaborative interactions.

This approach stems from our previous work in genetically diverse inbred mouse strains where millions of naturally occurring SNPs between strains were considered nature’s mutagenesis experiment (Heinz, Romanoski et al. Nature 2013; Gosselin et al., 2014). The current study is the first to implement this approach in human cells. The approach is as follows: SNPs are identified across the genome within a single individual. We achieved this using whole-genome sequencing (40X average depth) in one aortic endothelial cell donor. Next, we identified loci where the alleles of the SNPs mutated a canonical DNA binding motif for a TF of interest (AP1, ETS, or NFkB) only on one homologous chromosome. These are the potential functional SNPs that are used to test for effects on TF binding. Next, we developed an analytical pipeline using a combination of public tools (WASP, HOMER, R) to quantify allele-specific TF binding for JUN and NFkB/p65 that takes into account personal genome sequence and avoids caveats of mapping bias (described in the Materials and methods section).

Results show that allele-specific mutations in AP-1 motifs genome-wide (n = 237) are coincident with reduced JUN binding in the expected allele-specific manner (Figure 3, Figure 3—figure supplement 1 = 5e-10). This demonstrates that motif mutations have an effect on binding of the respective factor. Perhaps more interestingly, we found that ETS motif mutations were coincident with reduced JUN binding within 100 base pairs genome-wide (n = 150, p=3e-3). This provides further evidence that ETS and AP1 factors collaborate in binding to endothelial enhancers. We extend this approach to the later portion of our manuscript where we find that altered binding of NFkB/p65 associated with motif mutations in the kB motif itself as well as AP1 motifs and ET motifs (Figure 5—figure supplement 4). These data support a hierarchical role where AP1 and ETS factors prime the chromatin landscape for binding of signal-dependent factors like NFkB. These data and analytical details have been added to the manuscript as follows:

Results:

“Allele-specific binding to chromosomes lacking motif mutations supports collaborative binding between AP1 and ETS factors […] Taken together, these data support a collaborative relationship between AP1 and ETS factors at endothelial enhancers.”

Results: “In addition, we analyzed allele-specific NFkB binding as a function of motif mutations in the kB motif itself as well as mutations in the AP1 and ETS motif. […]Furthermore, these data demonstrate that allele-specific binding in heterogeneous human cells is a useful means to reveal collaborative interactions between transcription factors.”

Materials and methods:

“Whole-Genome Sequencing (WGS), Motif Mutation Analysis and Allele-Specific Factor Binding For the HAEC line sequenced for analysis as in Figure 3, genomic DNA was prepared for paired-end WGS on Illumina HiSeqX (Illumina, CA) by Novogene (Sacramento, CA) according to manufacturer specifications[…]This is a conservative approach, as many peaks have zero reads on the mutated chromosome.”

[Editors' note: further revisions were requested prior to acceptance, as described below.]

*The manuscript has been improved but there are some remaining issues that need to be addressed before acceptance, as outlined below:*

*There is one remaining point you may want to discuss. While the AP1 factors are highly expressed, the ETS factors are not. Figure 4—figure supplement 1 shows ERG expression to be less than 2.5 FPKM. Your data shows ERG and JUN co-binding on most regions. Is the protein of ERG that stable? Are the stoichiometry relationships reasonable?*

When quantifying gene expression from RNA-seq data as all tags between the transcription start site to the termination site, the expression levels between ETS and AP-1 factors are different. This is largely because ETS factors tend to have multiple large introns (e.g., ERG in Figure 1) and AP-1 factors lack introns (e.g., JUN in Figure 1). In both cases, however, the tag counts over the exons is comparable. This is further shown by comparison of microarray data measuring ETS and AP-1 factors (Figure 2), which shows similar levels of expression between ERG and JUN.

To enable more accurate comparisons between genes with different numbers of introns, we have now updated the definition by which RNA-seq data is counted (now restricted only to exons). This revision is reflected in Figure 1, Figure 1—figure supplement 1, Figure 4, and Figure 4—figure supplement 1. The results remain largely the same with the main exception that ERG is rank ordered higher among transcription factors in general (Figure 1). The results and conclusions of ERG knock-down experiments remain the same (Figure 4, and Figure 4—figure supplement 1).

While our study highlights a collaborative role for ERG and JUN specifically, we acknowledge that this combination is not likely the sole modulating combination. In fact, other collaborative interactions involving other AP-1 and ETS factors almost certainly regulate EC gene expression, and the cistromes of these factors and their functional effects remain a target of future investigation.

We have now revised our manuscript to reflect these changes in the following ways:

Discussion:

“While we focus on ERG and JUN in particular, other combinatorial interactions between other ETS and AP-1 members almost certainly play prominent roles in endothelial regulation.”

*Clearly, at their mRNA levels ERG and JUN are at very different, is there evidence that ERG is a more stable protein? A related point: In Figure 3—figure supplement 2, Two of the three examples (ERK and TEK) have very small fold changes in siERG. What about other genes that show stronger effects?*

We believe this comment refers to loci shown for TEK and CDH5. PECAM1, what was more significantly affected by siERG, is now included (Figure 3—figure supplement 2) that displays many ERG/JUN co-bound enhancers.

[Editors' note: further revisions were requested prior to acceptance, as described below.]

*The manuscript has been improved but there is one remaining issue that need to be addressed before acceptance, as outlined below:*

*While the methods for quantification of RNA-Seq data have been clearly improved since the original submission, there was consensus that these methods are not adequately described in the revised manuscript. Please expand and refine this description in the methods section.*

We now present a more detailed description of the RNA-seq analysis that reflects the precise options used to quantify gene expression and determine differential expression. This change can be found in subsection “RNA-seq Analysis” as follows:

“RNA-seq Analysis For quantification, RNA-seq was normalized using Reads Per Kilobase of transcript per Million mapped reads (RPKM) procedure in HOMER[…] Pathway enrichment analysis was performed using Ingenuity Pathway Analysis software (Qiagen).”